# Enhancing cross-protection against influenza by heterologous sequential immunization with mRNA LNP and protein nanoparticle vaccines

Chunhong Dong [1], Wandi Zhu[1], Lai Wei[1], Joo Kyung Kim[1], Yao Ma[1], Sang-Moo Kang [1] & Bao-Zhong Wang [1] ✉

Enhancing influenza vaccine cross-protection is imperative to alleviate the significant public health burden of influenza. Heterologous sequential immunization may synergize diverse vaccine formulations and routes to improve vaccine potency and breadth. Here we investigate the effects of immunization strategies on the generation of cross-protective immune responses in female Balb/c mice, utilizing mRNA lipid nanoparticle (LNP) and protein-based PHC nanoparticle vaccines targeting influenza hemagglutinin. Our findings emphasize the crucial role of priming vaccination in shaping Th bias and immunodominance hierarchies. mRNA LNP prime favors Th1-leaning responses, while PHC prime elicits Th2-skewing responses. We demonstrate that cellular and mucosal immune responses are pivotal correlates of cross-protection against influenza. Notably, intranasal PHC immunization outperforms its intramuscular counterpart in inducing mucosal immunity and conferring cross-protection. Sequential mRNA LNP prime and intranasal PHC boost demonstrate optimal cross-protection against antigenically drifted and shifted influenza strains. Our study offers valuable insights into tailoring immunization strategies to optimize influenza vaccine effectiveness.

Reoccurring seasonal influenza epidemics and occasional pandemics pose substantial threats to public health. The Centers for Disease Control and Prevention (CDC) of the United States recommends annual influenza vaccination as the most cost-effective and important prevention measure for individuals aged ≥6 months. Despite this, current seasonal influenza vaccines typically elicit strain-specific and short-lived immunity[1], offer limited cross-protection against antigenically diverse virus variants, and provide no defense against sporadic influenza pandemics. Influenza vaccine effectiveness (VE) has been suboptimal, ranging from 19% to 60%, depending on the degrees of antigenic divergence and prediction accuracy in different flu seasons[2]. Developing effective influenza vaccines or vaccination

strategies that can confer cross-protection against variant influenza viruses becomes a high priority to mitigate the public health consequences of influenza.

Including conserved antigens is a common strategy in the development of universal influenza vaccines[3,4], and some pioneering candidates have advanced to clinical trials[5]. However, the protection efficacy of these vaccines is not ideal due to the inherently poor immunogenicity of the highly conserved antigens. To date, no universal influenza vaccine has been approved for human use[6]. Mounting evidence suggests that heterologous sequential immunization strategies represent a promising alternative to inducing broad immunity. Heterologous sequential immunization may synergize diverse vaccine formulations

[1]Center for Inflammation, Immunity & Infection, Georgia State University Institute for Biomedical Sciences, 100 Piedmont Ave SE, Atlanta, GA 30303, USA.
✉ e-mail: bwang23@gsu.edu

and administration routes to induce multifaceted immune responses, thereby enhancing the protection potency and breadth[7,8].

Different vaccine formulations can elicit distinct immune responses depending on antigen types, delivery platforms, and administration routes[4,9]. Protein-based subunit vaccines typically evoke strong antibody responses but poor cellular responses[10]. In contrast, mRNA lipid nanoparticles (LNP) can trigger robust T-cell responses owing to an endogenous antigen production mechanism[11]. Additionally, mucosal immunity can be efficiently provoked by mucosal immunization but rarely by systemic vaccination like the intramuscular or intravenous routes[12,13]. As the first line of defense at the virus entry portal to prevent infections and transmissions, mucosal immunity has been recognized to be essential for preventing respiratory diseases and immune correlates of cross-protection against influenza[14,15]. Recent studies suggest that heterologous sequential immunization involving diverse routes holds promise in boosting both systemic and mucosal immunity. SARS-CoV-2 spike protein or adenoviral vector-based mucosal vaccine boosters following mRNA vaccination showed promising results in augmenting mucosal responses and combatting SARS-CoV-2 infections[13,16]. Nonetheless, many questions remain regarding the potential of tuning immunization strategies to broaden influenza vaccine protection and the identification of the immune correlates of cross-protection against influenza.

Here, we investigate if and how various prime-boost immunization strategies affect cross-protection efficacy against influenza using mRNA LNP and protein-based PHC nanoparticle vaccines targeting hemagglutinin (HA). In contrast to prior studies that utilized heterologous immunization to bolster homologous protection against SARS-CoV-2[12,13,16], our research delves into tailoring immunization strategies for influenza vaccines. Specifically, we emphasize achieving cross-protection against heterologous and heterosubtypic influenza strains and innovatively examine the impacts of various immunization combinations and sequences. Our findings underscore the importance of immunization orders and the crucial role of priming vaccination in shaping Th bias and immunodominance hierarchies of the immune responses. We also demonstrate that cellular and mucosal immune responses are important correlates of cross-protection against influenza in the absence of serum antibody cross-neutralization. Moreover, mucosal PHC nanoparticle immunization significantly outperforms its intramuscular counterpart in inducing mucosal immune responses and conferring cross-protection. Heterologous sequential mRNA LNP priming followed by intranasal protein nanoparticle boosting immunization confers optimal cross-protection against heterologous and heterosubtypic virus challenges. Our study offers valuable insights into customizing heterologous sequential immunization strategies to enhance vaccine efficacy and broaden protection.

## Results

### Priming vaccination plays critical roles in shaping Th bias and immunodominance hierarchies

Previously, we reported two influenza vaccines targeting full-length A/Aichi/2/1968 (Aic, H3N2) HA (H3) but in different formats—a recombinant protein-based polyethyleneimine-HA/CpG (PHC) nanoparticle and an HA mRNA LNP vaccine[17,18]. Here we fabricated H3-based PHC and mRNA LNP nanoparticles, with respective hydrodynamic sizes of 128.5 ± 2.207 nm and 86.59 ± 1.808 nm, along with polydispersity index values of 0.097 ± 0.073 and 0.111 ± 0.023, respectively (Supplementary Fig. 1a). We immunized mice and compared the serum antibody responses induced by one-dose PHC or mRNA LNP administered via intranasal (IN) and intramuscular (IM) routes, respectively. The two vaccines generated comparable H3-specific total IgG levels, indicating comparable antigen doses. However, they exhibited distinct antibody subtype profiles, suggesting different immunogenic properties (Supplementary Fig. 1b–d). mRNA LNP IM vaccination generated Th1-dominant antibody responses characterized by significantly higher

levels of IgG2a compared to IgG1. In contrast, PHC IN immunization generated Th2-skewing antibody responses (IgG1 > IgG2a) (Supplementary Fig. 1e).

To compare the vaccine immunogenicity through different immunization strategies, we primed and boosted mice with either PHC IN or mRNA LNP IM vaccination (Fig. 1a). Compared with single-dose PHC, mRNA LNP- and PHC-primed groups showed significant increases in H3-specific IgG, IgG1, and IgG2a antibody titers following the PHC booster (Fig. 1b). Notably, both the one- and two-dose PHC immunization groups showed a Th2-leaning antibody response (IgG1 > IgG2a), while the heterologous sequential IM mRNA priming-IN PHC boosting immunization, defined as IM (mRNA)+IN (PHC), displayed a Th1-leaning antibody profile (IgG1 < IgG2a), resembling the pattern observed with mRNA vaccination (Fig. 1c). Similar results were observed regarding how priming vaccination influenced mRNA LNP vaccination. Both mRNA- or PHC-primed groups displayed significantly higher antibody levels following the mRNA booster compared to single-dose mRNA vaccination (Fig. 1d). The sole mRNA immunization induced Th1-skewing responses, while sequential IN PHC priming-IM mRNA boosting immunization, defined as IN (PHC)+IM (mRNA), facilitated Th2-leaning responses, resembling the PHC vaccination pattern (Fig. 1e). Therefore, priming vaccination plays a significant role in shaping Th1/Th2 immune responses and antibody isotype production.

We further evaluated serum antibody cross-reactivity in different immunization groups. H3-specific IgG antibody responses were shown in Supplementary Fig. 2a as a comparison. Notably, serology using the vaccine antigen Aic H3 did not reflect cross-reactive antibody levels against variant influenza strains or trimeric full-length HAs. Diverse immunization strategies led to distinct immunodominance hierarchies, related to the Th1/Th2 antibody balance. Groups with Th2-skewing responses tended to have better IgG antibody cross-reactivity against distant strains. Among all groups, IM (mRNA)+IN (PHC) induced the highest IgG levels against H3 and homologous Aic (Supplementary Fig. 2a, b). However, two-dose IN (PHC)+IN (PHC) elicited the highest cross-reactive antibody levels against heterologous A/Philippines/2/1982 (Phi, H3N2) (Fig. 1f), A/Wisconsin/15/2009 (Wis, H3N2) (Supplementary Fig. 2c), heterosubtypic reassortant A/Shanghai/2/2013 (rSH, H7N9) (Fig. 1g) and A/Anhui/1/2013 (Anh, H7N9) HA (Supplementary Fig. 2d). Additionally, IN (PHC)+IM (mRNA) produced higher cross-reactive IgG against Wis, rSH, and Anh H7, despite lower H3- and Aic-specific IgG levels compared to the IM (mRNA)+IN (PHC) regimen. These results implied a possible correlation between the IgG subtype and cross-reactivity.

### Heterologous sequential IM (mRNA)+IN (PHC) immunization conferred optimal cross-protection

Following the analysis of antibody responses, we assessed the cross-protection efficacies in these groups against antigenically drifted and shifted influenza viruses. Previously, we have demonstrated that both PHC and mRNA vaccines conferred complete homologous protection in a two-dose regimen in mice, yielding 100% survival rates and no discernible bodyweight loss, even under high challenge doses of 15 ~ 30× $LD_{50}$ Aic[17,18].

Mouse-adapted heterologous Phi (H3N2) was employed to evaluate protection against antigenically drifted viruses four weeks post-boost. Upon infection, naive mice lost body weight rapidly and succumbed within 8 days (Fig. 2a, b). Both the IM (mRNA)+IN (PHC) and IN (PHC)+IN(PHC) groups demonstrated 100% survival rates and complete protection against bodyweight loss. However, the IN (PHC)+IM (mRNA) and IM (mRNA)+IM (mRNA) groups experienced significant bodyweight losses of up to 12.1% and 11.2%, respectively. One out of the five mice in the IN (PHC)+IM (mRNA) group approached the humane endpoint on day 5 post-challenge. Moreover, the IM (mRNA)+IN (PHC) and IN(PHC)+IN(PHC) groups exhibited significantly higher 8-day body weight

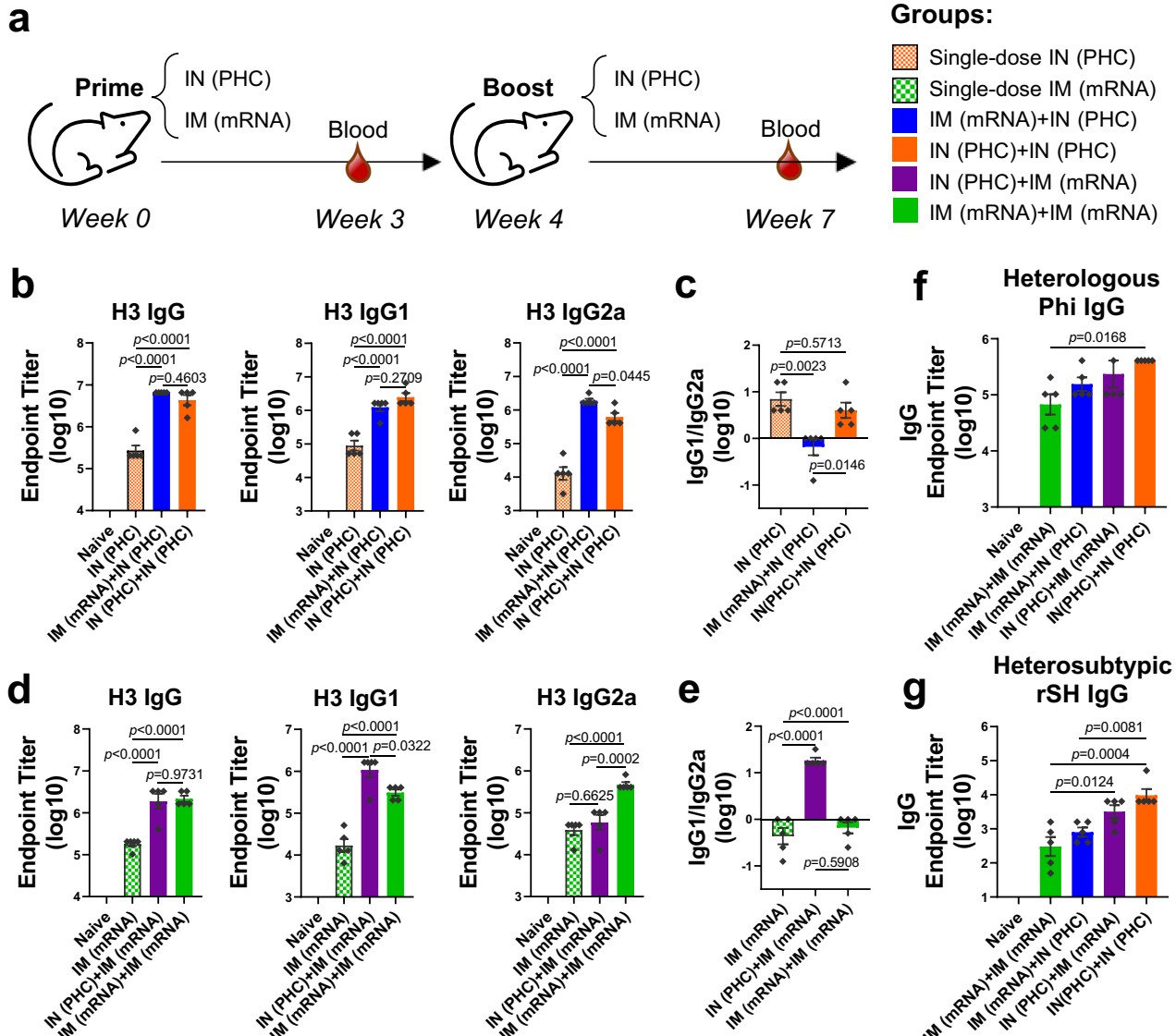

**Fig. 1 | The influence of priming vaccination on the antigen-specific IgG1/IgG2a antibody responses and antibody cross-reactivity. a** Schematic diagram of immunization and sera collection. Mice were immunized twice with either IM (mRNA) or IN (PHC) vaccines at an interval of 4 weeks. Sera were collected 3 weeks post-prime for one-dose IN (PHC) and IM (mRNA) groups or 3 weeks post-boost for two-dose groups. **b, c** Comparison of the antigen-specific total IgG, IgG subtype (IgG1 and IgG2a) levels, and IgG1/IgG2a ratio in IN (PHC), IM (mRNA)+IN (PHC), and IN (PHC)+IN (PHC) groups. **d, e** Comparison of the antigen-specific total IgG, IgG subtype (IgG1 and IgG2a), and IgG1/IgG2a ratio in IM (mRNA), IN (PHC)+IM (mRNA), and IM (mRNA)+IM (mRNA) groups. **f, g** Cross-reactive IgG antibody levels against heterologous Phi and heterosubtypic rSH viruses. IM intramuscular, IN intranasal, PHC polyethyleneimine-HA/CpG, Phi A/Philippines/2/1982 (H3N2), rSH reassortant A/Shanghai/2/2013 (H7N9). Group definitions: Single-dose IN (PHC), orange with patterns, Single-dose IM (mRNA), green with patterns, IM (mRNA)+IN (PHC), blue; IN (PHC)+IN (PHC), orange; IN (PHC)+IM (mRNA), purple; IM (mRNA)+IM (mRNA), green. Data are presented as mean ± SEM ($n = 5$). Statistical significance was analyzed by one-way ANOVA followed by Turkey's multiple comparison tests. Source data are provided as a Source Data file.

under the curve (AUC) than IN (PHC)+IM (mRNA) and IM (mRNA)+IM (mRNA) groups. We further determined Phi-specific IgG isotype levels by ELISA assay and observed similar results with that of H3 (Fig. 2c, d). PHC-primed groups showed significantly higher IgG1/IgG2a ratios versus mRNA-primed groups. The IN(PHC)+IN(PHC) displayed the highest IgG1, while IM (mRNA)+IN (PHC) exhibited the highest IgG2a levels among all groups. However, no neutralization activity against Phi was observed across all groups (Supplementary Fig. 2e).

The SH (H7N9), a strain within influenza A phylogenetic group 2 like Aic, originates from poultry and carries pandemic potential. It has caused high-mortality human infections in eastern China since 2013. We challenged mice with the heterosubtypic mouse-adapted rSH to evaluate protection efficacies against antigenically shifted viruses. We excluded the IM (mRNA)+IM (mRNA) group due to its unsatisfactory

heterologous protection but retained the IN (PHC)+IM (mRNA) group for comparison with the IM (mRNA)+IN (PHC) group. Among all groups, the IM (mRNA)+IN (PHC) group exhibited the best protection along with the least bodyweight loss post-rSH infection (Fig. 2e, f), despite suboptimal levels of rSH-specific antibodies (Fig. 1g). This group displayed significantly higher 8-day and 14-day bodyweight AUC than all the other groups. The IN (PHC)+IN (PHC) showed a higher 14-day bodyweight AUC than the naive and IN (PHC)+IM (mRNA) groups. All immunized mouse groups showed 100% survival rates (Fig. 2g). Further analysis revealed no serum neutralization activity against rSH, with an IgG1-dominant H7-specific antibody profile consistently observed across all groups (Supplementary Fig. 2f, g). This antibody profile provides additional support for the hypothesis regarding the correlation between IgG subtype and cross-reactivity against distant strains.

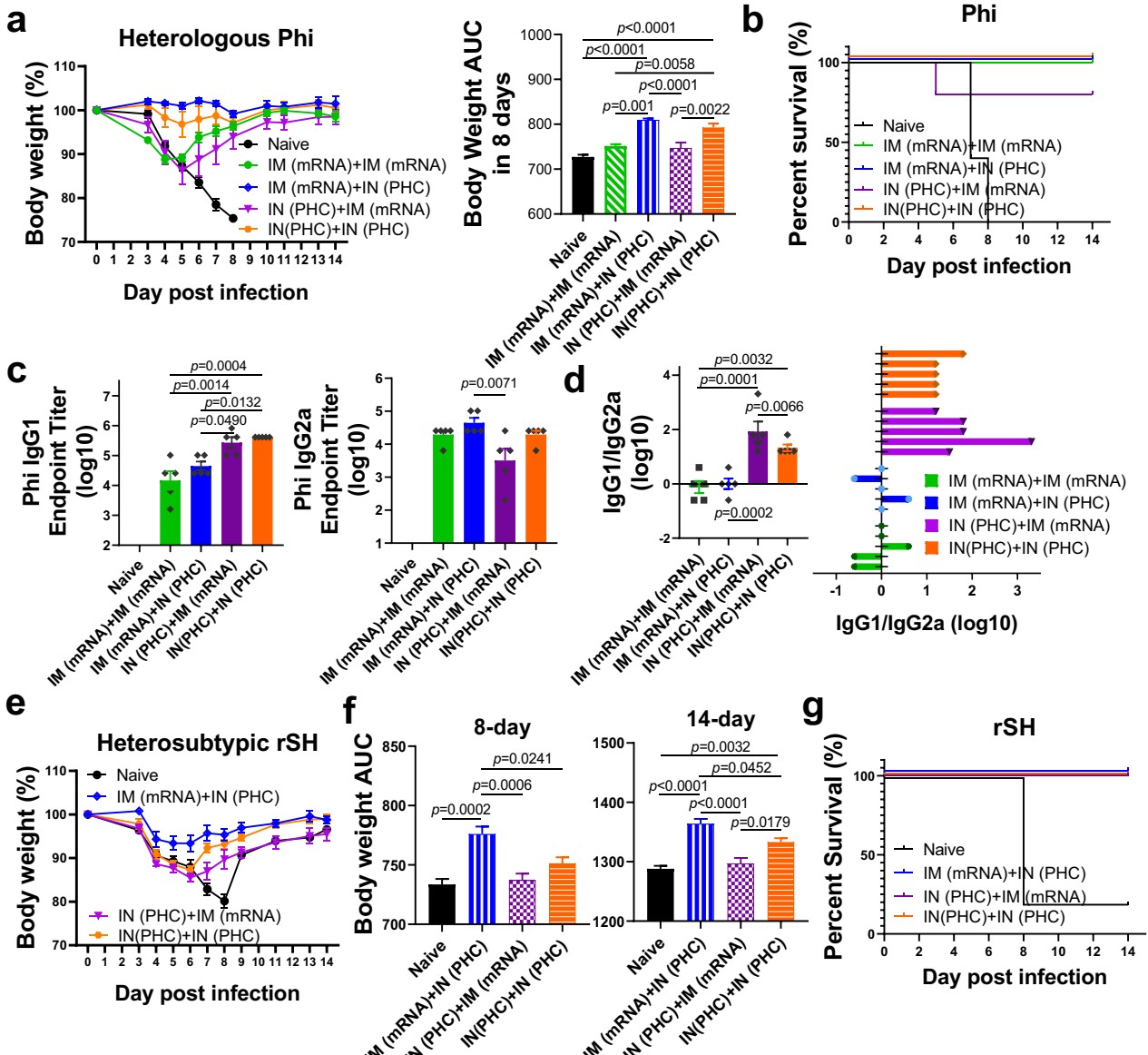

**Fig. 2 | Cross-protection efficacies against heterologous and heterosubtypic strains. a** Mouse body weight changes and 8-day body weight AUC post heterologous Phi H3N2 challenge. **b** Mouse mortality post-Phi challenge. **c** Phi-specific IgG1 and IgG2a levels post-boosting immunization. **d** The calculated IgG1/IgG2a ratios (log10) in immune sera. **e, f** Mice body weight changes, and 8-, and 14-day body weight AUC post heterosubtypic rSH H7N9 virus challenge. **g** Mouse mortality post-rSH challenge. The body weight AUC and its standard error for each group in **a** and **f** were calculated by GraphPad Prism 8. Phi A/Philippines/2/1982 (H3N2), rSH reassortant A/Shanghai/2/2013 (H7N9), AUC the area under the curve. Group definitions: naive, black; IM (mRNA)+IM (mRNA), green; IM (mRNA)+IN (PHC), blue; IN (PHC)+IM (mRNA), purple; IN (PHC)+IN (PHC), orange. Data are presented as mean ± SEM ($n = 5$). Statistical significance was analyzed by one-way ANOVA followed by Turkey's multiple comparison tests. Source data are provided as a Source Data file.

Therefore, two-dose Aic H3 mRNA LNP IM vaccination proves inadequate for heterologous protection against Phi. The heterologous sequential IM (mRNA)+IN (PHC) immunization demonstrated the most robust cross-protection against both heterologous and heterosubtypic virus challenges. Moreover, despite lower serum antibody cross-reactivity, IM (mRNA)+IN (PHC) surpassed the reversed IN (PHC)+IM (mRNA) in conferring cross-protection. This underscores the importance of immunization orders and the potential contributions of immune responses beyond serum antibodies.

## IM (mRNA)+IN (PHC) immunization elicited the most robust and balanced T-cell responses

Cellular immune responses play important roles in controlling viral infection by producing effector cytokines and eliminating virus-infected cells[19,20]. We evaluated cytokine-secreting splenocytes by cytokine ELISpot assay 5 weeks post-boosting immunization. One-dose IN (PHC) immunization was included for comparison. IL-2 is a T-cell growth factor essential for T-cell proliferation and the generation of effector and memory cells[21]. IFN-γ and IL-4, potent inducers and indicators of Th1 and Th2 immune responses, respectively, are associated with distinct IgG antibody subclasses and cellular responses[22,23]. We found that two-dose vaccination induced significantly more IL-2-, IFN-γ-, and IL-4-secreting splenocytes than naive and one-dose PHC group (Fig. 3a–c), aligning with the elevated antibody levels in two-dose versus one-dose groups (Fig. 1). Moreover, among all groups, the heterologous sequential IM (mRNA)+IN (PHC) immunization demonstrated the largest IL-2- and IFN-γ-secreting splenocyte populations. However, no significant

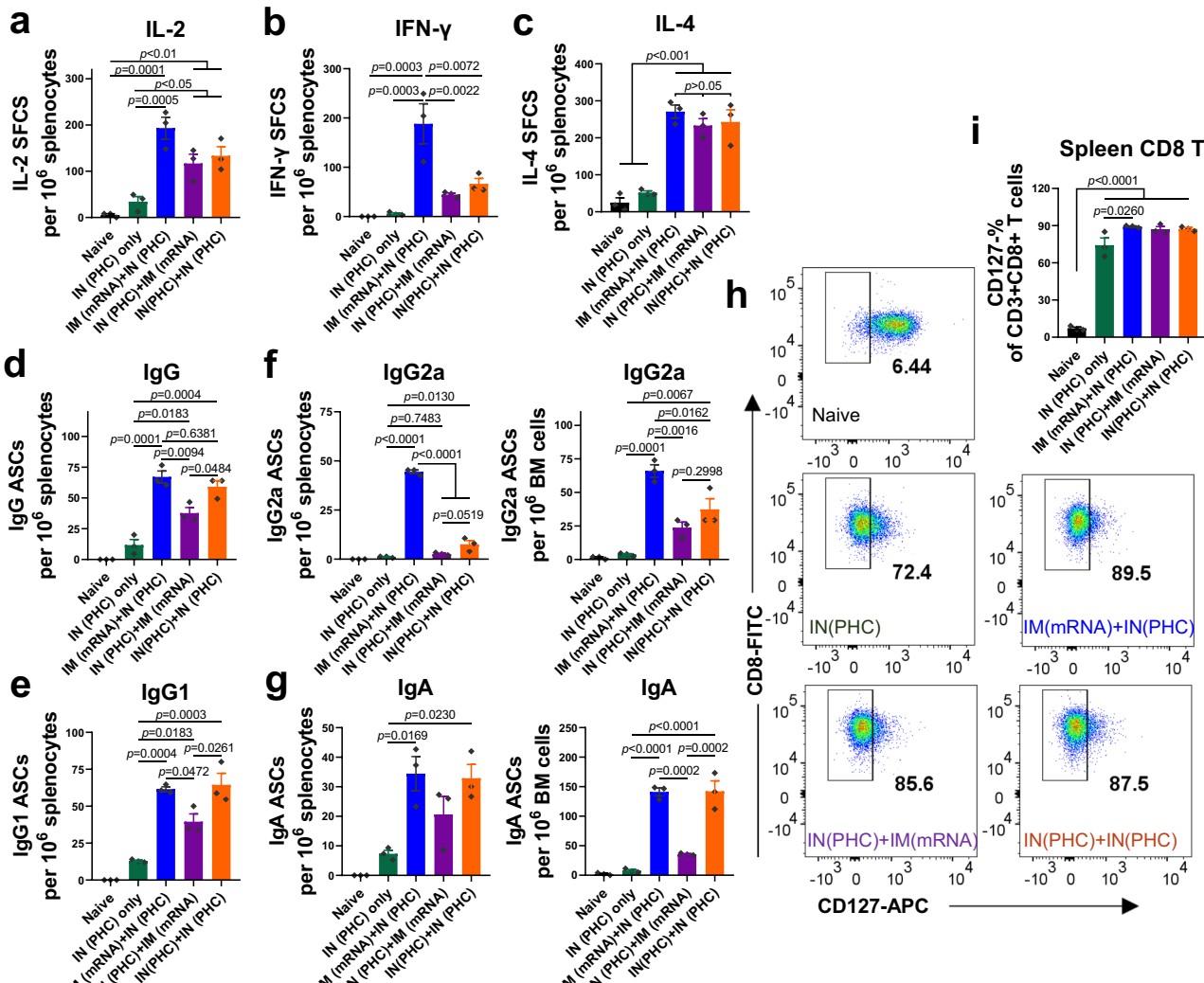

**Fig. 3 | Cellular immune responses. a–c** IL-2-, IFN-γ and IL-4-secreting splenocytes. **d, e** IgG and IgG1 antibody-secreting cells in mouse spleens. **f, g** IgG2a and IgA ASCs in mouse spleens and bone marrow, respectively. **h, i** Downregulation of CD127 on CD8+ T cells in mouse spleens after 2-day antigen-re-stimulation. Spleens and bone marrow were collected 5 weeks post-boosting immunization. ASC antibody-secreting cells, SFC spot-forming cells, BM bone marrow. Group definitions: naive, black; IN (PHC) only, dark green; IM (mRNA)+IN (PHC), blue; IN (PHC)+IM (mRNA), purple; IN (PHC)+IN (PHC), orange. Data are presented as mean ± SEM (*n* = 3). Statistical significance was analyzed by one-way ANOVA followed by Turkey's multiple comparison tests. Source data are provided as a Source Data file.

difference was observed in IL-4-secreting cell frequencies among the two-dose groups. Therefore, the heterologous IM (mRNA)+IN (PHC) elicited the most robust and balanced Th1 and Th2 immune responses.

We also evaluated antibody-secreting cell (ASC) frequencies in mouse spleens. Naive mice did not have any antigen-specific ASCs. Two-dose vaccinations induced significantly more antigen-specific IgG and IgG1 ASCs than the one-dose PHC group (Fig. 3d, e). Specifically, the heterologous sequential IM (mRNA)+IN (PHC) immunization generated IgG and IgG1 ASC frequencies comparable to IN (PHC)+IN (PHC), but significantly higher than the reversed IN (PHC)+IM (mRNA) immunization, underscoring the effects of immunization orders. The heterologous sequential IM (mRNA)+IN (PHC) immunization elicited the most abundant IgG2a ASCs in both spleens and bone marrows (Fig. 3f), while the reversed IN (PHC)+IM (mRNA) and homologous IN (PHC)+IN (PHC) produced low IgG2a ASC levels, consistent with the Th-leaning of the antibody responses. Additionally, IM (mRNA)+IN (PHC) and IN (PHC)+IN (PHC) produced more IgA ASCs than other groups (Fig. 3g).

We detected antigen-responsive T cells using an ex vitro antigenic re-stimulation method. The IL-7/IL-7R (CD127) pathway plays an important role in regulating T-cell homeostasis by modulating CD4+ and CD8+ T-cell generation[24,25]. In addition to the existence on naive T cells, CD127 has been identified as a selective characteristic marker for long-lived memory T cells and undergoes down-regulation upon contact with antigens[26], distinguishing long-living memory T cells from effector T cells. After a 2-day re-stimulation, the immunized groups displayed a significant down-regulation of CD127 on both CD8+ (Fig. 3h, i) and CD4+ (Supplementary Fig. 3) T cells compared to the naive group. Besides, IM (mRNA)+IN (PHC) exhibited enhanced CD127 down-regulation on CD8+ T cells versus the one-dose PHC group.

Therefore, two-dose sequential immunizations not only enhanced antibody responses but also bolstered cellular responses, including cytokine-secreting T cells and antibody-secreting B cell populations, compared to a single-dose PHC. Notably, the heterologous sequential IM (mRNA)+IN (PHC) regimen elicited the most robust and balanced T-cell responses, particularly with significantly stronger IFN-γ-mediated cellular responses than all other groups.

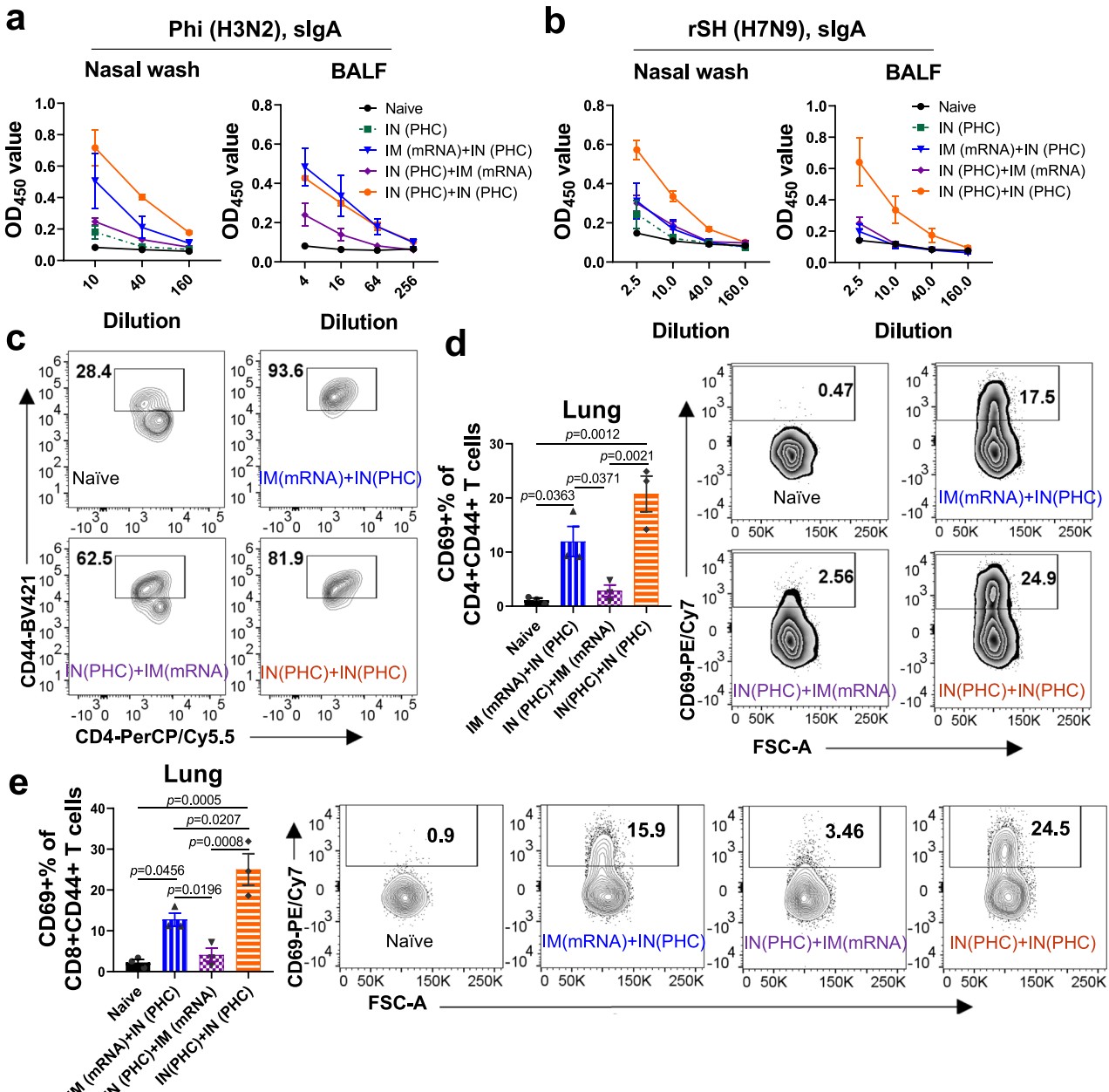

**Fig. 4 | Adaptive immune responses at mucosal surfaces and in pulmonary tissues. a, b** Cross-reactive sIgA antibody levels against Phi and rSH in nasal washes and BALF, respectively. **c** Comparison of CD4+CD44+ populations in BALF. **d** Comparison of CD4+CD44+CD69+ T-cell populations in mouse lung. **e** Comparison of CD8+CD44+CD69+ T-cell populations in mouse lung. Phi A/Philippines/2/1982 (H3N2), rSH reassortant A/Shanghai/2/2013 (H7N9), sIgA secretory IgA, BALF bronchoalveolar lavage fluid. Group definitions: naive, black; IN (PHC) only, dark green; IM (mRNA)+IN (PHC), blue; IN (PHC)+IM (mRNA), purple; IN (PHC) +IN (PHC), orange. Data are presented as mean ± SEM (*n* = 3). Statistical significance in **d** and **e** was analyzed by Student's *t* test (IM (mRNA)+IN (PHC) vs. IN (PHC)+IM (mRNA)) or one-way ANOVA followed by Turkey's multiple comparison tests (between other groups). Source data are provided as a Source Data file.

## Adaptive immune responses at mucosal surfaces and in pulmonary tissues

Mucosal vaccination can evoke adaptive mucosal immunity, including secretory IgA (sIgA) and tissue-resident memory T and B-cell responses. sIgA, the principal antibody subtype at respiratory mucosal surfaces, typically exhibits superior cross-reactivity compared to systemic IgA antibodies owing to their polymeric structure[27]. sIgA has been demonstrated protective against many respiratory virus infections, including SARS-CoV and influenza[28,29].

We collected nasal washes and bronchoalveolar lavage fluid (BALF) 5 weeks post-boost immunization and determined cross-reactive sIgA antibody levels. The two-dose immunization groups showed hierarchical H3-specific sIgA levels in mucosal washes depending on the immunization routes and orders: IN (PHC)+IN (PHC) > IM (mRNA)+IN (PHC) > IN (PHC)+IM (mRNA), consistent with the IgA levels in the boost sera (Supplementary Fig. 4a). The IN (PHC) +IN (PHC) group showed optimal performance in producing cross-reactive sIgA antibodies in both nasal washes and BALF. The heterologous sequential IM (mRNA)+IN (PHC) immunization generated higher sIgA levels against H3, Aic, Phi, Wis, and A/mallard/Netherlands/ 1/1999 HA (Net H4) than the reversed IN (PHC)+IM (mRNA) regimen (Fig. 4a and Supplementary Fig. 4a–d). However, minimal differences were observed in sIgA levels against rSH (Fig. 4b) and Anh H7 (Supplementary Fig. 4e) between these two groups, probably attributed to

the low sIgA levels resulting from substantial genetic divergence between these strains and Aic. Additionally, two-dose IN (PHC) immunization induced significantly higher sIgA levels against all tested viruses or HAs in nasal washes than one-dose PHC, underscoring the enhancing effects of boosting vaccinations on sIgA production at mucosal surfaces. A separate study displayed similar boosting effects in BALF (Supplementary Fig. 4f). Therefore, intranasal immunization can effectively evoke sIgA production at mucosal surfaces, and two-dose IN (PHC) outperformed the one-dose regimen. In heterologous immunization strategies, the sequential IM (mRNA)+IN (PHC) regimen demonstrated superiority over the reversed IN (PHC)+IM (mRNA) regimen.

It has been reported that sIgA dominates the antibody response in the upper respiratory tract. In contrast, IgG is the major antibody isotype in the lower respiratory tract. Therefore, we further analyzed the antigen-specific IgG in BALF. No significant difference was observed in H3- and Phi-specific IgG titers among the indicated two-dose immunization groups (Supplementary Fig. 4g, h). However, the IM (mRNA)+IN (PHC) group exhibited Th1-leaning subtype profiles, whereas IN (PHC)+IM (mRNA) and IN (PHC)+IN (PHC) displayed Th2-leaning profiles. Meanwhile, the IN (PHC)+IN (PHC) group elicited the most robust Anh H7-specific IgG responses in BALF, dominated by IgG1, among all groups (Supplementary Fig. 4i). These results were consistent with our observations in sera. Additionally, no discernible neutralization activity against either Phi or rSH viruses was detected in BALF (Supplementary Fig. 4j, k).

Growing evidence indicates that memory T and B cells resident along the respiratory tract are highly effective at providing rapid and potent protection against respiratory viral infections such as influenza and SARS-CoV-2[12,30]. We collected BALF 5 weeks post-boost immunization and analyzed airway T-cell populations. Few immune cells were detected in the BALF from naive mice. A marked increase of CD45+ and antigen-experienced (CD44+) CD4+ lymphocytes was observed in the immunized versus naive group (Fig. 4c, Supplementary Fig. 5a). The heterologous sequential IM (mRNA)+IN (PHC) immunization group displayed CD45+ and CD4+CD44+ cell populations comparable to the IN (PHC)+IN (PHC) group but significantly higher than the IN (PHC)+IM (mRNA) group. Thus, intranasal boosting immunization can substantially promote airway CD45+ and CD4+CD44+ T lymphocyte accumulation which may act immediately upon virus infection. The heterologous sequential IM (mRNA)+IN (PHC) immunization outperformed the reversed IN (PHC)+IM (mRNA) regimen in the induction of these crucial cells and elicited responses comparable to those of two-dose IN (PHC) immunizations.

Studies suggested that not all memory T-cell subsets are equally protective. Tissue-resident memory T cells ($T_{RM}$) are particularly potent in protecting against subsequent heterologous influenza infections[31,32]. CD69 is an important distinguishing cell-surface marker constitutively expressed on CD4 and CD8 $T_{RM}$ cells under steady-state conditions in mouse lung tissues. It functions as a critical antagonist of S1PR1 (CD363), limiting cell egress into the bloodstream[30,33]. CD69, along with CD44, are useful for studying mouse $T_{RM}$. We analyzed tissue-resident memory cells in mouse lungs and observed significantly boosted CD4+CD44+CD69+ and CD8+CD44+CD69+ cells from the IM (mRNA)+IN (PHC) and IN (PHC)+IN (PHC) groups versus the naive and IN (PHC)+IM (mRNA) groups (Fig. 4d, e, Supplementary Fig. 5b). Moreover, we studied lung-resident B memory cells ($B_{RM}$) by analyzing the expression of classical memory B cell marker CD38 and tissue retention marker CD69. The IM (mRNA)+IN (PHC) immunized mice displayed a significant increase in CD19+IgD−CD38+CD69+ $B_{RM}$ cell populations compared to naive mice (Supplementary Fig. 5c). Hence, the heterologous sequential IM (mRNA)+IN (PHC) regimen facilitated the generation of lung-resident memory T and B cells, potentially shielding the mice from diverse virus infections.

## Mucosal immunization outperformed intramuscular immunization in cross-protection potency

Mucosal immunity likely plays a crucial role in cross-protection, as indicated by our analysis results of diverse immune responses. To evaluate the contributions of mucosal immune responses to cross-protection, we compared the IM (mRNA)+IN (PHC) to the IM (mRNA)+IM (PHC) group. Upon heterologous Phi challenge, all mice from the IM (mRNA)+IM (PHC) group displayed noticeable body weight drops, while IM (mRNA)+IN (PHC) mice did not show apparent body weight loss, although both groups survived Phi challenges (Fig. 5a, Supplementary Fig. 6a, b). The heterologous sequential IM (mRNA)+IN (PHC) immunization group demonstrated significantly higher body weight AUC than IM (mRNA)+IM (PHC) in 8 (Fig. 5b) and 14 days (Supplementary Fig. 6a) post-Phi infection.

Analysis of the immune sera indicated that IM (mRNA)+IN (PHC) immunization induced significant Phi-specific sIgA in nasal washes and BALF, whereas IM (mRNA)+IM (PHC) did not induce this response (Fig. 5c). However, comparable Phi- and/or H3-specific IgG antibody profiles were observed in immune sera and BALF between these two groups (Supplementary Fig. 6c−f). We compared the antigen-specific IgG1/IgG2a ratios to assess whether the immunization routes impacted the Th1/Th2 immune balance depicted in Fig. 1. Our results indicated comparable Phi- and H3-specific IgG1/IgG2a ratios between the two groups. Furthermore, no neutralization activity against Phi was detected in either sera or BALF in both groups (Supplementary Fig. 6g).

Also, upon heterosubtypic rSH challenge, the IM (mRNA)+IN (PHC) group showed better protection than the IM (mRNA)+IM (PHC) group, as evidenced by less body weight loss and quicker recovery (Fig. 5d, Supplementary Fig. 6h, i) and significantly higher body weight AUC (Fig. 5e) in 14 days post-infection, despite comparable rSH-specific IgG levels in immune sera (Supplementary Fig. 6j). Similarly, no neutralizing activity against rSH was detected (Supplementary Fig. 6k). Significantly boosted rSH-specific mucosal sIgA were observed in the IM (mRNA)+IN (PHC) group compared to the naive and IM (mRNA)+IM (PHC) groups (Fig. 5f). Although IM (mRNA)+IM (PHC) induced a low-level H3-specific serum IgA, it failed to elicit mucosal sIgA against H3, Aic, Wis, Net H4, or A/mallard/Sweden/51/2002 HA (Swe H10) (Supplementary Fig. 7a−f). Thus, intranasal, rather than intramuscular, PHC-boosting immunization in a sequential immunization strategy effectively induces sIgA at mucosal surfaces.

Moreover, IM (mRNA)+IN (PHC) elicited higher levels of airway CD45+ lymphocytes, CD4+CD44+T cells, CD8+CD44+ T cells, lung-resident CD4+CD44+CD69+ and CD8+CD44+CD69+ $T_{RM}$ cells, and CD19+IgD−CD38+CD69+ $B_{RM}$ cells than naive and IM (mRNA)+IM (PHC) groups, while no significant difference was observed between IM (mRNA)+IM (PHC) and naive group (Fig. 5g−i, Supplementary Fig. 7g).

We further compared the homologous IN (PHC)+IN (PHC) versus IM (PHC)+IM (PHC) regimen upon 3× $LD_{50}$ Phi challenge. The IN (PHC)+IN (PHC) group showed robust protection with no noticeable body weight loss post-infection, while IM (PHC)+IM (PHC) displayed body weight drops of up to 9.4% (Fig. 5j, Supplementary Fig. 8a). The IN (PHC)+IN (PHC) group displayed significantly higher 14-day body weight AUC than the IM (PHC)+IM (PHC) group. However, no difference in antigen-specific IgG, IgG1, and IgG2a levels, as well as IgG1/IgG2a ratios was observed in immune sera between these two groups (Supplementary Fig. 8b−e). The comparable IgG1/IgG2a ratios indicated that it was the priming vaccine itself that significantly shaped the Th immune balance and the production of antibody isotypes, rather than the administration routes.

Meanwhile, we observed that one-dose IN (PHC) protected mice with a 100% survival rate but suffered from severe body weight loss (Fig. 5j, Supplementary Fig. 8a). Thus, two vaccination doses are required for optimal cross-protection. Therefore, despite comparable

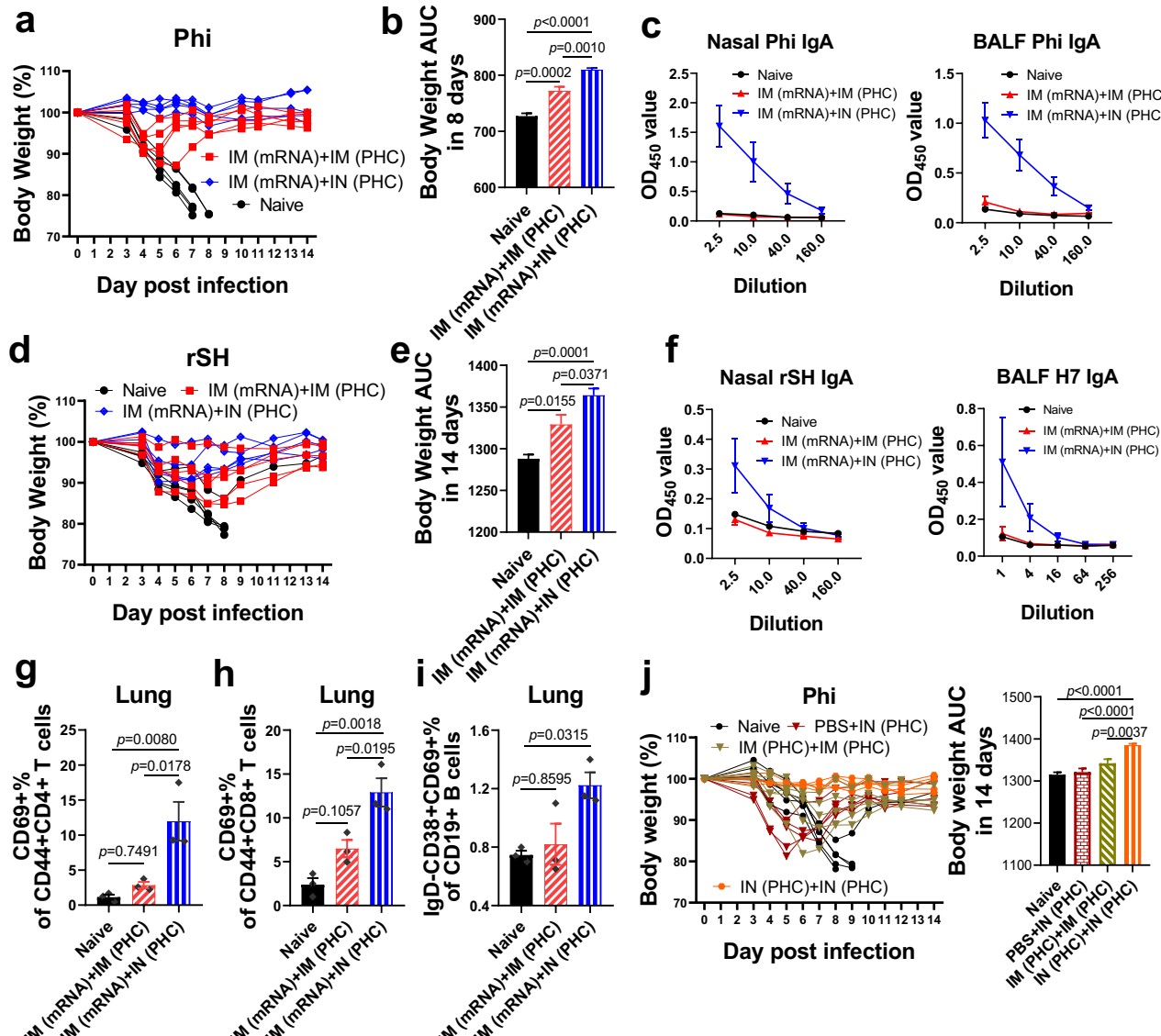

**Fig. 5 | Intranasal immunization outperformed intramuscular regimens in cross-protection efficacy. a, b** Mouse body weight changes and 8-day bodyweight AUC post-Phi challenge in IM (mRNA)+IN (PHC) and IM (mRNA)+IM (PHC) groups. **c** Phi-specific sIgA levels in nasal washes and BALF. **d, e** Mouse body weight changes and 14-day bodyweight AUC post-rSH challenge in IM (mRNA)+IN (PHC) and IM (mRNA)+IM (PHC) groups. **f** rSH-specific sIgA levels in nasal washes and BALF. **g, h** The localized CD44+CD69+ T-cell populations in mouse lungs. **i** Lung-resident memory B cell populations. **j** Mouse body weight changes and 14-day bodyweight AUC post-Phi challenge in PBS+IN(PHC), IN (PHC)+IN (PHC), and IM (PHC)+IM

(PHC) groups. Samples in **c**, **f**, and **g–i** were collected 5 weeks post-boost. Phi A/Philippines/2/1982 (H3N2), rSH reassortant A/Shanghai/2/2013 (H7N9), AUC the area under the curve, BALF bronchoalveolar lavage fluid. Group definitions: naive, black, IM (mRNA)+IN (PHC), blue, IM (mRNA)+IM (PHC), red; PBS+IN (PHC), maroon; IM (PHC)+IM (PHC), khaki green; IN (PHC)+IN (PHC), orange. Data in histograms are presented as mean ± SEM ($n = 5$ for **a**, **b** and **d**, **e**, and **j**; $n = 3$ for **c** and **f**–**i**). The body weight AUC and its standard error in **b**, **e**, **j** were calculated by GraphPad Prism 8. Statistical significance was analyzed by one-way ANOVA followed by Turkey's multiple comparison tests. Source data are provided as a Source Data file.

capability to induce systemic antibody responses, intranasal immunization outperformed its intramuscular counterparts, whether in homologous or heterologous immunization regimens, in providing influenza cross-protection.

**mRNA-LNP vaccine serves as an excellent priming modality for intranasal protein-based vaccine boosters**
We have demonstrated that IM (mRNA) priming played critical roles in modulating the immune responses. The heterologous sequential IM (mRNA)+IN (PHC) exhibited the highest efficacy in cross-protection. To further assess the significance of mRNA-LNP vaccine priming, we conducted a group involving intramuscular mRNA-LNP priming followed by an intranasal soluble HA booster. It is worth noting that

soluble HA is poorly immunogenic when administered intranasally and cannot confer cross-protection against heterologous Phi in a two-dose immunization regimen[17].

We compared the protective efficacies of the IM (mRNA)+IN (H3) and IM (mRNA)+IN (PHC) groups against a 3× LD$_{50}$ heterologous Phi challenge (Supplementary Fig. 9a, b). Our findings revealed that administering a soluble H3 intranasal booster following mRNA priming also provided robust heterologous protection. Moreover, we observed no significant difference in the 14-day body weight AUC between these two groups. This result further strengthens the importance of mRNA as the priming vaccination modality. Additionally, the IM (mRNA)+IN (H3) group induced Th1-leaning antibody responses (IgG1 < IgG2a) comparable to those of the IM (mRNA)+IN (PHC) group. These

responses differed from those observed with H3 immunization[17] but were similar to mRNA-LNP, further affirming the importance of priming vaccination (Supplementary Fig. 9c–f).

## Discussion

Influenza poses a substantial burden on public health, resulting in significant mortality and morbidity worldwide annually. Seasonal influenza VE fluctuates and is often unsatisfactory, notably dropping to as low as 19% for the 2014–2015 flu season due to the significant drift of circulating A/H3N2 influenza viruses[34]. Enhancing the potency of influenza vaccine cross-protection has become an imperative task. Heterologous immunizations with diverse vaccine types have the potential to bolster vaccine cross-protection by eliciting more diverse and multifaceted immune responses. Since the onset of the COVID-19 pandemic, advances in mRNA-LNP vaccine development have diversified heterologous vaccination formats.

HA, the most abundant influenza virus surface glycoprotein, serves as the most potent immunogen and primary target for influenza vaccine development. Previously, we reported two vaccines targeting Aic HA: an mRNA LNP or a PHC protein nanoparticle vaccine, both demonstrating robust homologous protection[17,18]. Here, we compared the immunogenicity of IM mRNA LNP and IN PHC vaccines. It's important to note that a direct comparison between mRNA and protein immunization requires caution. In the present study, we observed comparable levels of antigen-specific IgG antibodies with the indicated dose, suggesting comparable antigen doses. Additionally, the distinct Th1- and Th2-leaning immune characteristics reflected their separate mRNA or protein vaccine features. Then, we investigated how different immunization strategies−homologous or heterologous−impact the ultimate cross-protection efficacy using these two vaccines. We utilized Phi and rSH as challenge strains to assess the cross-protection efficacies against antigenically drifted and shifted influenza viruses. The heterologous sequential IM (mRNA)+IN (PHC) immunization demonstrated the best cross-protection against both Phi and rSH infections, resulting in the least bodyweight loss among all groups. Two-dose PHC vaccination conferred good protection against Phi, consistent with our prior findings[17], but showed less efficacy against rSH. By contrast, the reversed IN (PHC)+IM (mRNA) and two-dose IM mRNA groups experienced severe body weight loss. We also observed similar suboptimal heterologous protection against Phi in two-dose mRNA vaccination even when adjuvanted by α-GC or cGAMP[18].

To determine the key correlates of cross-protection, we thoroughly investigated the antibody, cellular, and mucosal responses. Despite superior cross-protection and significant IgG against homologous Aic or H3, the heterologous sequential IM (mRNA)+IN (PHC) immunization generated fewer cross-reactive antibodies, particularly those against rSH, when compared with the reversed IN (PHC)+IM (mRNA) or the homologous two-dose IN (PHC)+IN (PHC) regimen. Hence, cross-reactive serum antibodies were not the primary correlates of the cross-protection. Further analysis revealed the absence of cross-neutralizing activity in the immune sera against Phi[17] and rSH. Our findings also indicated the crucial role of priming vaccination in shaping Th bias and immunodominance hierarchies of the immune responses. In a sequential immunization, PHC prime promoted Th2-leaning responses, while mRNA LNP prime facilitated Th1-leaning responses. Further analysis indicated that variations in the administration route (IM or IN) for PHC immunization did not lead to a significant change in the IgG1/IgG2a ratios. Therefore, it appears that the vaccine itself, administered during the priming vaccination, significantly shapes the Th immune balance and antibody isotype production, rather than the administration routes. Potential underlying mechanisms may involve the immune milieu, such as Th1/Th2 CD4 T cells and the cytokine environment following priming immunization[35].

Our results also revealed a potential correlation between antibody Th profiles and cross-reactivity contingent upon the antigenic divergence distance. We observed similar Th profiles against homologous H3 and drifted Phi virus. In a sequential vaccination, PHC prime facilitated significantly higher Phi-specific IgG1/IgG2a ratios than mRNA prime, consistent with observations against H3. Heterologous sequential IM (mRNA)+IN (PHC) immunization elicited balanced Phi-specific IgG1 and IgG2a antibodies. However, all immunization groups displayed IgG1-dominant profiles against antigenically shifted Anh H7. Enhanced IgG antibody cross-reactivity against distant rSH and Anh H7 was noted in groups with Th2-skewing responses. Nonetheless, further evidence is needed to confirm this speculation. Considering the IgG subtype profiles, IgG2a-mediated effector functions[36,37], such as antibody-dependent cellular cytotoxicity and phagocytosis (ADCC, ADCP), may play some roles in protection against Phi, albeit to a lesser extent against rSH.

Cellular immunity plays a vital role in limiting disease severity and has been recognized as an important correlate of cross-protection against influenza due to its broad cross-reactivity[38,39]. Surprisingly, significant CD127 down-regulation on splenic T cells upon antigen-restimulation in vitro was observed across all immunization groups compared to the naive group, including the one-dose PHC immunization. Multiple markers may be required for the precise characterization of long-lived memory T cells in the future. However, our results indicated that heterologous sequential IM (mRNA)+IN (PHC) immunization elicited the most robust and balanced Th1/Th2 cytokine-secreting cell and IgG1/IgG2a ASC responses. IFN-γ plays an essential role in protective cellular immunity[40]. Effector CD8 T cells produce cytokines such as IFN-γ to limit early virus replication and directly kill target cells, promoting efficient virus elimination[41,42]. Effector CD4 cells expressing IFN-γ and perforin were reported to have cytolytic activity and mediate quick recovery from influenza infection[43]. Heterologous sequential IM (mRNA)+IN (PHC) immunization induced the highest IFN-γ- and IL-2-secreting but comparable IL-4-secreting splenocyte frequencies versus other immunization groups. Moreover, cytokine expression was reported to influence the avidity of the cytotoxic T lymphocytes (CTL)−Th1 cytokine IL-2 facilitates high-avidity CTL generation, while Th2 cytokine IL-4 negatively affects CTL functional avidity[44,45]. Considering the distinct cytokine secretion profiles, IFN-γ-mediated effector functions and the avidity of T cells probably contribute to the high cross-protection efficacy in the IM (mRNA)+IN (PHC) group.

Mucosal immunity including sIgA and $T_{RM}$ also correlates with cross-protection. Our findings revealed elevated mucosal sIgA levels in nasal washes and BALF from groups receiving mucosal boosters compared to those undergoing the mucosal prime-systemic boost regimen. However, the IgG profiles in BALF mirrored those in the serum, consistent with the previous report[1]. Moreover, no evident neutralization activity was detected in BALF across all groups. Meanwhile, IM (mRNA)+IN (PHC) and IN (PHC)+IN (PHC) elicited enhanced airway CD45+ and antigen-experienced CD4+CD44+ T lymphocyte accumulation, along with more robust localized lung-resident CD4+CD44+CD69+ and CD8+CD44+CD69+ $T_{RM}$ versus IN (PHC)+IM (mRNA). A recent study suggests that systemic immunity can transition into local immunity following a mucosal booster[16], potentially explaining the robust mucosal and cellular immunity observed in the IM (mRNA)+IN (PHC) group. We further compared the different immunization routes to evaluate the contribution of mucosal immunity. Our results demonstrated that the induction of mucosal immune responses was primarily determined by the immunization route rather than the type of vaccine. Repeated intramuscular vaccination did not stimulate effective mucosal immunity despite high serum antibody responses and mice receiving two-dose IM mRNA LNP or PHC vaccine formulations experienced severe bodyweight loss post-Phi or rSH infection. Mucosal immunization outperformed intramuscular

immunization in conferring cross-protection. Therefore, the robust protective mucosal immunity accounts for the significant cross-protection in the IM (mRNA)+IN (PHC) and IN (PHC)+IN (PHC) groups observed in our study.

To sum up, our study highlights the importance of immunization orders and the crucial role of priming vaccination in steering antibody Th1/Th2 bias and shaping immunodominance hierarchies in a sequential immunization. We further showed that cross-reactive antibodies without cross-neutralization activity in immune sera are not a predictive correlate of cross-protection. By contrast, cellular and mucosal immune responses are more important correlates of cross-protection. Our findings emphasize the pivotal role of mucosal immunity in influenza cross-protection. Mucosal immunity can act as the first line of defense to immediately recognize and eliminate viruses at the site of entry, potentially preventing virus infections and transmissions[16,46]. In contrast, circulating memory cells require time to proliferate, migrate, and mediate effector function[47]. While serum antibodies remain the most accessible and convenient indicators for predicting protection, they may not capture the full picture and complexity of immunity or accurately reflect the breadth of protection against variant strains. Evaluating influenza vaccine cross-protection should involve mucosal and cellular immunity, along with profiling antibody subtypes. However, the challenge of detecting these responses in human populations, especially localized T cell and antibody responses, remains a significant limitation. Consequently, more effective detection or analysis methods are warranted.

Despite the lipid toxicity of current mRNA-LNP vaccines hindering their application in mucosal routes to achieve effective mucosal immunity[48], in our study, the mRNA-LNP vaccine proved its worth as an excellent priming vaccination modality, particularly when followed by intranasal boosters, eliciting comprehensive correlates of cross-protection. Soluble HA intranasal booster following HA mRNA priming also offered good heterologous protection, which was consistent with earlier results of COVID-19 vaccines from the Iwasaki Group[16]. We assume that seasonal influenza vaccines are probably applicable as mucosal boosters, and live attenuated influenza vaccine (LAIV), viral vector-based, and mRNA vaccine platforms that can induce robust Th1 immunity are potentially promising priming modalities.

Our findings underscore the advantages of heterologous sequential immunizations combining diverse vaccine formulations and administration routes, along with customized immunization sequences. A tailored immunization strategy can improve the magnitude, quality, and diversity of the adaptive immune responses in several aspects: (1) synergizing different immunization routes and vaccine types to elicit multifaceted immune responses; (2) customizing priming modality to influence the Th bias and promote robust Th1-skewing cross-protective cellular responses; (3) leveraging systemic responses during priming to bolster mucosal responses following booster immunizations. These enhanced cross-protective cellular and mucosal immune responses contributed to influenza cross-protection in the absence of antibody neutralization. Tailoring the immunization strategy to elicit multifaceted cross-protective cellular and mucosal immune responses with diverse vaccine formulations appears to be a viable alternative to the development of broad-spectrum influenza vaccines.

We have presented that the sequential mRNA LNP prime plus mucosal PHC nanoparticle boost strategy achieved robust systemic and mucosal immunity and provided potent cross-protection against antigenically drifted and shifted influenza variants within phylogenetic Group 2, with Aic HA as the target antigen. Quadrivalent vaccine formulations containing two influenza A and two influenza B strains hold promise for universal protection against recurring flu epidemics and preparedness against future pandemics. Our research contributes valuable insights into customizing immunization strategies to enhance vaccine efficacy and expand the scope of protection. These findings

are instrumental in advancing the clinical translation and adoption of more effective sequential immunization strategies. Leveraging the expedited progress of vaccine development, future investigations should prioritize cross-protective influenza vaccines or vaccination strategies to combat consistent influenza viral mutations[49].

## Methods

### Study design
The main aim of this study was to investigate the impact of immunization strategies on the induction of cross-protective immune responses and identify the most effective strategy for robust and broad influenza protection. To this end, we conducted various homologous or heterologous prime-boost immunizations utilizing mRNA LNP and protein-based PHC influenza vaccines, examined their induced antibody, cellular, and mucosal responses, and investigated how they correlate with the cross-protection efficacies against antigenically drifted and shifted influenza strains in mice. We further evaluated the contributions of mucosal responses to cross-protection. The number of mice per experimental group was indicated in the figure legends. Statistical analyses were conducted when applicable.

### Ethics statement
The entire study was approved by Georgia State University Institutional Animal Care and Use Committee (IACUC). All mouse experiments were performed in strict compliance with the IACUC guidelines of Georgia State University under IACUC protocol A22029. Female Balb/c mice (6–8 weeks old) were purchased from Envigo and housed at Georgia State University at 20–23 °C, 45–55% relative humidity with 12-hour light/dark cycles, free food, and water supplies. Female animals were chosen because males are aggressive and often fight and get injured, which interferes with data collection. Mice were adapted for no less than one week before experiments and were grouped randomly.

### Proteins and viruses
The recombinant GCN4-stabilized trimeric full-length influenza HA ectodomains (Aic H3, Net H4, and Swe H10) composing both the head and the stalk regions were expressed by the Bac-to-Bac baculovirus expression system (Thermo Fisher Scientific) and purified from the baculovirus-infected Spodoptera frugiperda 9 (Sf9, American Type Culture Collection (ATCC), CRL-1711) cell cultures using HisPur Ni-NTA resins (Catalog No.: 88223, Thermo Scientific) according to the manufacture's instructions. The recombinant H3 derived from the HA gene of A/Aichi/2/1968(H3N2) (GenBank No: CY121117.1) was designed as previously described[50,51]. The Net H4 and Swe H10 constructs were generated using the full-length HA plasmids derived from A/mallard/Netherlands/1/1999 (H4N6) (Catalog No.: NR-28996) and A/mallard/Sweden/51/2002 (H10N2) (Catalog No.: NR-29002) obtained from BEI resources. The purified protein purity was verified by reducing sodium dodecyl sulfate-polyacrylamide gel electrophoresis (SDS-PAGE) followed by Coomassie Blue staining and visualization with ChemiDoc Touch imaging system (Bio-Rad, USA). The A/Anhui/1/2013 HA (Anh H7, NR-44365, and NR-44081) was obtained from BEI resources.

The Aic, Phi, Wis, and reassortant rSH H7N9 influenza viruses were expanded in embryonated chicken eggs[52]. The reassortant rSH viruses were generated as previously described[51]. The Phi and rSH virus strains used for challenge studies were mouse-adapted. The $LD_{50}$ (median lethal dose) of influenza viruses was determined by the standard Reed and Muench method.

### mRNA-LNP and PHC nanoparticle vaccine production
The H3 mRNA production and mRNA-LNP fabrication processes have been described in our previous study[18]. The H3 coding sequence was derived from the HA gene of A/Aichi/2/1968(H3N2) (GenBank No: CY121117.1) and provided in the Supplementary text. Briefly, mRNAs

were produced using a MEGAscript™ T7 Transcription Kit and purified using a MEGAclear™ Transcription Clean-Up Kit from Invitrogen. To prepare the mRNA-LNP nanoparticles, mRNA in 25 mM sodium citrate buffer (pH 4.0) was formulated with a lipid mixture in ethanol at a mass ratio of 1:20, and a volume ratio of 3:1 using a microfluidic mixer (Precision Nanosystems). The lipids contain DOTMA, DOPE, cholesterol, and DMG-PEG 2000 at a molar ratio of 50:10:38.5:1.5. The PHC (PEI-H3/CpG) nanoparticles were prepared as described previously[17]. The recombinant H3 generated in our lab was formulated with PEI (Sigma-Aldrich, USA), and CpG ODN1826 (InvivoGen, USA) by a facile vortex-mixing method. The particle size was determined by dynamic light scattering (DLS) with a Malvern Zetasizer Nano ZS (Malvern Panalytical).

## Vaccination and sample collections

To compare the immunogenicity of the mRNA-LNP and PHC, female Balb/c mice (6–8 weeks, $n = 5$ per group) were intramuscularly (IM) injected with 50 μL of mRNA-LNPs containing 5 μg of Aic H3 mRNA in DPBS, or intranasally (IN) immunized with 30 μL of PHC nanoparticles containing 5 μg of H3 protein in DPBS once, respectively. Sera samples were collected three weeks after a single dose of immunization, referred to as prime sera, for antibody evaluation. Naive mice were used as controls.

To study the influence of immunization strategies, female Balb/c mice (6–8 weeks, $n = 5$ per group) were immunized twice with either IM mRNA LNP or IN PHC vaccines in a typical 'prime+boost' regimen at an interval of 4 weeks (Fig. 1a). Four sequential immunization strategies were included, including homologous IM (mRNA)+IM (mRNA) and IN (PHC)+IN (PHC), and heterologous IM (mRNA)+IN (PHC) and IN (PHC) +IM (mRNA). Sera samples were collected three weeks after the booster immunization, referred to as boost sera.

To evaluate the induction of cellular and mucosal immune responses, female Balb/c mice (6–8 weeks, $n = 3$ per group) were sacrificed five weeks post-boosting immunization. Spleens, bone marrow, and lungs were collected, processed into single-cell suspensions, treated with RBC Lysing Buffer Hybri-Max (Sigma-Aldrich, USA), and finally resuspended in complete RPMI medium (cRPMI, RPMI 1640 supplemented with 2 mM L-glutamine, 10% FCS (v/v), 1% (v/v) penicillin/streptomycin, 1× non-essential amino acids, 1 mM sodium pyruvate, 10 mM HEPES, and 50 μM β-mercaptoethanol). Mouse nasal washes and BALF were collected with 200 μl and 1.5 mL of ice-cold DPBS supplemented with 0.5 % BSA, respectively. The mucosal washes were centrifuged at 550 g for 5 minutes, and the resulting supernatant was stored at −20 °C until analysis. The cells in BALFs were pelleted down followed by treatment with RBC Lysing Buffer Hybri-Max and resuspension in complete RPMI medium.

## Challenge studies in Balb/c mice

To study the cross-protection efficacy of the indicated four sequential immunization strategies, the immunized female Balb/c mice (6–8 weeks, $n = 5$ per group) were intranasally infected with either mouse-adapted heterologous A/Philippines/2/1982 (Phi, H3N2, challenge dose: $2 \times LD_{50}$) or heterosubtypic A/Shanghai/2/2013 (rSH, H7N9, challenge dose: $3 \times LD_{50}$) viruses, at 4 weeks post-boosting immunization. Naive mice were used as controls. Mouse body weight changes were recorded daily for 2 weeks post-infection. A weight loss of >20% was used as a humane endpoint. The body weight area under the curve (AUC) for each group was calculated from the body weight curve by GraphPad Prism v8.0.

We further evaluated the cross-protection efficacies of the same vaccine formulations administered through different routes. The vaccination doses remain consistent with the amount mentioned above. 50 or 30 μL of vaccine suspensions were used for intramuscular injection or intranasal instillation, respectively. For the comparison of IM (mRNA)+IM (PHC) versus IM (mRNA)+IN (PHC), female Balb/c mice (6–8 weeks, $n = 5$ per group) were primed with IM (mRNA) and subsequently boosted with either IM (PHC) or IN (PHC) following the same immunization schedules as mentioned above. Mice were then challenged with $2 \times LD_{50}$ of Phi or $3 \times LD_{50}$ of rSH viruses 4 weeks post-boost immunization. For the comparison of IM (PHC)+IM (PHC) versus IN (PHC)+IN (PHC), female Balb/c mice (6–8 weeks, $n = 5$ per group) were immunized intramuscularly or intranasally with two doses of PHC nanoparticles at an interval of 4 weeks. Mice were then challenged with $3 \times LD_{50}$ of Phi viruses 4 weeks post-boost immunization. One-dose IN (PHC) vaccination was included for comparison.

To assess the significance of mRNA-LNP vaccine priming, we compared the protective efficacies of the IM (mRNA)+IN (H3) and IM (mRNA)+IN (PHC) groups. Female Balb/c mice (6–8 weeks, $n = 5$ per group) were primed intramuscularly with 5 μg of mRNA LNPs and subsequently boosted with either IN (H3) or IN (PHC) containing 5 μg of H3. Mice were then challenged with $3 \times LD_{50}$ of Phi viruses as described above.

## Enzyme-linked immunosorbent assay (ELISA) assay

ELISA assay was performed to evaluate antigen-specific antibody responses in immune sera and mucosal washes, as previously described[53]. Briefly, 96-well ELISA plates (Thermo Scientific, USA) were coated with purified recombinant proteins or whole formalin-inactivated viruses (4 μg/mL, 50 μL/well) overnight at 4 °C, washed with PBST (PBS with 0.05% Tween-20), and blocked with PBST supplemented with 2% BSA for 1 h at 37 °C. Serial dilutions of immune serum, nasal washes, or BALF were added to the plates followed by incubation at 37 °C for 2 h. After three washes with PBST, 50 μL of horseradish peroxidase (HRP)-conjugated goat anti-mouse IgG (Cat: 1033-05, Lot: J3316-P623D, SouthernBiotech, USA), IgG1 (Cat: 1071-05, Lot: K1619-N733, SouthernBiotech, USA), IgG2a (Cat: 1080-05, Lot: B4520-RC63B, SouthernBiotech, USA), or IgA (Cat:1040-05, Lot: J4416-M729, SouthernBiotech, USA) antibodies were added to the plates at a dilution ranging from 1:2000 to 1:4000 and incubated at 37 °C for 1 h. The plates were washed, and 50 μL of 3,3′,5,5′-tetramethylbenzidine (TMB) was added to the plates as chromogenic substrates, followed by the addition of 50 μL of 1 M $H_2SO_4$ as the stop solution. The absorbance values at 450 nm were recorded using Biotek Epoch Microplate Reader (Agilent). The highest dilution with an optical density at 450 nm ($OD_{450}$) twice that of the naive mouse sample was used as the endpoint antibody titer. The IgG isotype (IgG1 and IgG2a) levels were evaluated to calculate the ratio of IgG1/IgG2a titers and determine the Th1/Th2 bias of the antibody responses.

To study the vaccine immunogenicity, the antibody levels against the vaccine antigen (H3) and homologous Aic virus were determined. To evaluate the antibody cross-reactivity, Phi H3N2, A/Wisconsin/15/2009 (Wis, H3N2), rSH H7N9, Anh H7, Net H4, and Swe H10 were used as coating antigens to determine the specific IgG levels. To evaluate the mucosal antibody responses, sIgA levels in nasal washes and BALF were determined.

## Microneutralization assay

Microneutralization assays were performed to evaluate antibody neutralization activity against Phi and rSH in immune sera and BALF according to the WHO protocol for Serological Diagnosis of Influenza. The $TCID_{50}$ (median tissue culture infectious dose) of Phi and rSH virus was determined in Madin−Darby Canine Kidney (MDCK (NBL-2), ATCC CCL-34) cells by the Reed and Muench method. MDCK cells were purchased from ATCC and maintained in the lab as recommended by the vendor. Briefly, two-fold serial dilutions of BALF or receptor-destroying enzyme (Denka Seiken Co., Ltd)-treated and heat-inactivated immune sera were mixed with 100 $TCID_{50}$ of Phi or rSH virus in the presence of 2 μg/mL of TBCK-trypsin for 1 h incubation at 37 °C. The mixture was added to MDCK cells ($1.5 \times 10^4$ cells/well) and incubated overnight. A standard influenza nucleoprotein-based ELISA

assay was used to determine virus inhibition. Mouse anti-influenza A virus NP antibody (Clone: C43, Ab128193, abcam) was used at a dilution of 1:2000. Immune sera from 5 mice were pooled, and then triplicate samples were tested.

## B cell enzyme-linked immunospot (ELISpot) assay

B cell ELISpot assays were performed to evaluate antigen-specific ASC frequencies as previously described[28]. Briefly, purified H3 (4 μg/mL, 50 μL/well) was pre-coated on sterile 96-well ELISpot filtration plates (Millipore, USA) overnight at 4 °C. The plates were washed with PBS, blocked with culture medium for 2 h at 37 °C, and then $5 \times 10^5$ splenocytes or bone marrow cells were seeded per well and incubated for 16 h at 37 °C. After removing cells and washing, HRP-conjugated goat anti-mouse IgG, IgG1, IgG2a, or IgA antibodies were added at a dilution of 1:1000 to the plates followed by incubation for 1 h at room temperature. Following five washes, KPL True Blue Peroxidase substrate (SeraCare) was added to develop spots. The reaction was terminated with rinsing water. Results were recorded with BIOSYS Bioreader-6000-E (BioSystem).

## T-cell ELISpot assay

T-cell ELISpot assays were performed to analyze antigen-specific cytokine (IL-2, IFN-γ, or IL-4)-secreting cell frequencies in mouse spleens collected 5 weeks post-boosting immunization. Briefly, sterile 96-well ELISpot filtration plates (Millipore, USA) were pretreated with anti-mouse IL-2 (Clone: JES6-1A12, Cat: 503704, Biolegend), IFN-γ (Clone: R4-6A2, Cat: 551216, BD Biosciences), or IL-4 (Clone: 11B11, Cat: 504102, Biolegend) capture antibodies (4 μg/mL, 50 μL/well) overnight at 4 °C. After washing and blocking, $5 \times 10^5$ splenocytes were seeded per well and cultured for 2 days at 37 °C with 4 μg/mL of H3 as stimuli. After removing cells, plates were incubated with 50 μL of biotin-conjugated detection antibodies, anti-mouse IL-2 (Clone: JES6-5H4, Cat: 503804, Biolegend), IFN-γ (Clone: XMG1.2, Cat: 554410, BD Biosciences), or IL-4 (Clone: BVD6-24G2, Cat: 504202, Biolegend), at a dilution of 1:500 at 37 °C for 1 h. After washing, streptavidin-HRP (Cat: 554066, BD Biosciences) was added for another 1 h incubation. KPL True Blue Peroxidase substrate was used to develop spots, and spots were recorded with BIOSYS Bioreader-6000-E.

## Flow cytometry

Flow cytometry was used to analyze the lymphocyte populations within immunized mouse spleens, BALF, and lungs collected 5 weeks post-boosting immunization. Antibodies were diluted at a dilution of 1:150 for cell staining.

Antigen-responsive T cells in immunized mouse spleens were evaluated by an ex-vitro antigenic re-stimulation experiment. $1 \times 10^6$ splenocytes were cultured in cRPMI medium in the presence of 4 μg/mL of H3 and 1ug/mL of anti-CD28 at 37 °C for 2 days for re-stimulation. The re-stimulated splenocytes were pelleted down and washed with FACS buffer (DPBS supplemented with 2% FCS). Then the cells were stained with an antibody cocktail containing anti-mouse CD3e-PE (Clone: 145-2c11, eBioscience, Cat: 12-0031-82), CD4-Percp/Cy5.5 (Clone: RM4-5, BD Pharmingen™, Cat:550954), CD8α-FITC (Clone: 53-6.7, Biolegend, Cat:100712), CD127-APC (Clone: A7R34, Biolegend, Cat:135011), CD16/32 (Clone: 2.4G2, BD Pharmingen™, Cat:553142) antibodies, and Zombie NIR viability dye (Biolegend, Cat:423105) for 30 min at 4 °C in dark. The gating strategy for the T-cell populations is displayed in Supplementary Fig. 3.

For BALF T-cell staining, the cells in BALFs were washed and stained with an antibody cocktail containing anti-mouse anti-mouse CD45-PE (Clone: 30-F11, Biolegend, Cat:103105), CD4-Percp/Cy5.5, CD8α-FITC, CD44-BV421 (Clone: IM7, Biolegend, Cat:103040), CD16/32 antibodies, and Zombie NIR viability dye for 30 min at 4 °C. The gating strategy for the T-cell populations is displayed in Supplementary Fig. 5a.

Mouse lung tissues were cut into small pieces and treated with 1 mg/mL Collagenase D (Roche) and 30 μg/mL DNase I (Sigma-Aldrich) in RPMI 1640 media for 30 min at 37 °C followed by grinding through a 70 μm cell strainer. The cells were pelleted down and then treated by RBC Lysing Buffer Hybri-Max to remove the red blood cells. After washing with the FACS buffer, one-third of the cells were stained with the T-cell antibody cocktail or the B-cell antibody cocktail in FACS buffer for 30 min at 4 °C. The lung T-cell antibody cocktail contains the following antibodies: anti-mouse CD45-PE, CD4-Percp/Cy5.5, CD8α-FITC, CD44-BV421, CD69-PE/Cy7 (Clone: H1.2F3, Biolegend, Cat:104511), CD16/32 antibodies, and Zombie NIR viability dye. The lung B cell antibody cocktail contains the following antibodies: anti-mouse CD19-APC (Clone: 1D3, BD Pharmingen™, Cat:550992), CD45R/B220-AF700 (Clone: RA3-6B2, Biolegend, Cat:103231), IgD-FITC (Clone: 11-26 c.2a, Biolegend, Cat:405703), CD69-PE/Cy7, CD38-Pacific Blue (Clone: 90, Biolegend, Cat:102719), CD16/32 antibodies, and Zombie NIR viability dye. The gating strategy for the lung cell populations is displayed in Supplementary Fig. 5b, c.

The stained cells were fixed with 2% paraformaldehyde for 15 min at 4 °C before analysis. Data were acquired on a BD LSRFortessa™ Cell Analyzer (BD Biosciences), and results were further analyzed by FlowJo v.10 software.

## Statistical analysis

Data are presented as mean ± standard error of the mean (SEM). Data plots/graphs were created by GraphPad Prism v8.0 (GraphPad Software). Statistical analyses were performed using one-way analysis of variance (ANOVA) followed by Turkey's or Dunnett's multiple comparison tests or unpaired two-tailed Student's $t$ test in GraphPad Prism v8.0 when applicable. A probability value of $p > 0.05$ is recognized as not significant. $p < 0.05$ is recognized as statistically significant. $p < 0.01$ was considered extremely significant.

## Reporting summary

Further information on research design is available in the Nature Portfolio Reporting Summary linked to this article.

## Data availability

All data generated in this study are provided in the paper and its Supplementary Information files. Source data are provided with this paper.

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

## Acknowledgements

This work was financially supported by the US National Institutes of Health (NIH)/National Institute of Allergy and Infectious Diseases (NIAID) under grants R01AI101047 and R01AI143844 to B.-Z.W.

## Author contributions

Conceptualization: C.D. Investigation, methodology, data analysis: C.D., W.Z., L.W., J.K., Y.M., S.-M.K., and B.-Z.W. Funding acquisition, Project administration, and Supervision: B.-Z.W. Writing—original draft: C.D. Writing—review & editing: C.D., W.Z., L.W., J.K., Y.M., S.-M.K., and B.-Z.W. All authors provided experimental materials and contributed to the final draft.

## Competing interests

The authors declare no competing interests.
