## [Peer Review File · Nature Communications]

Enhancing Cross-Protection against Influenza by Heterologous Sequential Immunization with mRNA LNP and Protein NanoparticlesReviewers' Comments:

Reviewer #1:

Remarks to the Author:

In this manuscript, Dong et al. investigated multiple homologous and heterologous sequential vaccination strategies in cross-reactive immunity against influenza. The authors demonstrated that 1) vaccine-induced immune responses exhibited Th1 and Th2 biases following mRNA-LNP and CpG-adjuvanted protein nanoparticle vaccine immunization, respectively, 2) intranasal (IN) route of immunization induced stronger mucosal immunity and cross-reactive protection than intramuscular (IM) route, and 3) IM mRNA-LNP priming followed by IN protein nanoparticle boosting afforded optimal cross-reactive infection. The manuscript provided interesting insights. The authors should be applauded for their comprehensive efforts in comparing the effect of immunization routes, vaccine platforms, and immunization orders on vaccine-induced influenza immunity. The ELISA data as well as the viral protection data were compelling. The analysis of cross-reactive immunity using heterologous and heterosubtypic flu strains was thorough. And the manuscript was well written and clearly presented. However, several issues reduced my enthusiasm for the paper (see below):

1. Similar strategies utilizing IM mRNA-LNP prime followed by IN boost with protein or adenoviral vector vaccines have been reported previously (Lapiente et al., *Nature Communications*, 2021; Mao et al., *Science*, 2022). In addition, the exact formulations of the vaccines have also been reported by the authors, reducing the novelty of the manuscript. While the authors have acknowledged these points in the paper, they should extensively highlight the substantial advancement this manuscript has made compared to previous studies, such as their creative exploration of different immunization combinations, the use of a different viral pathogen as well as the focus on the protective effects on heterologous and heterosubtypic strains.

2. Throughout the manuscript, the authors have compared the immunogenicity of IN PHC vs IM mRNA (either as a prime or a boost). An important conclusion that the authors made was that IM mRNA vs IN protein nanoparticle vaccinations induce distinct Th1/Th2 bias and isotype skewing. Several caveats to consider regarding this comparison:

- a. Direct comparison between mRNA and protein immunization warrants precaution since the antigen dose could not be controlled (5 ug mRNA may not be "translated" to 5 ug protein). Importantly, the authors showed that IM mRNA priming and IN PHC priming induced comparable titers of total H3 IgG (Fig S1B), suggesting that the antigen doses between the two vaccine platforms were similar. These points should be discussed in the manuscript.
- b. Adjuvants play an important part in shaping the antibody isotypes. For PHC vaccines, have the authors examined adjuvants other than CpG? In other words, is Th2 skewing an intrinsic property of PHC immunization or is it also dependent on adjuvant types?
- c. Additionally, have the authors looked at IM PHC immunization? Are the immune responses similarly Th2 biased?

3. The methodologies used in this manuscript to assess mucosal immunity were flawed, which made data interpretation difficult:

- a. The lung tissues are richly vascularized. This means that immune responses measured in the lung would be heavily "contaminated" with immune cells from the circulation if circulating cells are not distinguished (via intravenous CD45 labeling) or removed (through perfusion) from tissue-resident cells. Without a clear way of separating the two compartments, the conclusions about mucosal cellular immunity cannot be reliably drawn. This applies to measurement of both mucosal T cells and B cells.
- b. In addition, all mucosal cellular responses measured in the manuscript were polyclonal which made the data not particularly informative. Tetramer-based approaches should be used to identify antigen-specific T cells and B cells. Alternatively, ELISPOT experiments should be done for lung tissues if tetramer reagents are not available.
- c. Absolute cell count (# of antigen-experience CD4 and CD8 T cells in the lung and airways) should be included in addition to representative flow plots and relative percentages.

4. To further strengthen the serology data, serum and mucosal neutralization against homologous and heterologous influenza strains should be provided for each immunization strategies. In addition, mucosal IgG titer in the BALF and nasal washes should be measured. Do mucosal IgG titers track with protection?

5. A thorough discussion of why the heterologous sequential immunization strategy led to better cross-protection would further strengthen the manuscript.

Below I listed additional minor issues and comments I have that if addressed could further improve the quality of the manuscript.

6. In Line 91, besides hydrodynamic sizes (Fig S1A), polydispersity index (PDI) of PHC and LNP nanoparticles should be reported as well.

7. In Line 119, the findings that IgG1 leads to improved antibody cross-reactivity is interesting. Can the authors provide some potential mechanisms?

8. In Line 202, the authors performed ELISPOT assay to detect ASC response in the spleen and bone marrow. The bone marrow data are important and should be moved from Fig S3 in the main Figures.

9. In Line 446, results shown in Fig S9 should be described in the Main Text, not just in the Discussion section.

10. In Fig S7, error bars are missing.

Reviewer #2:

Remarks to the Author:

The authors have investigated the effect of a prime – boost regiment with intra nasal mRNA lipid nanoparticle and intra muscular protein-based PHC nanoparticle vaccines targeting influenza hemagglutinin, H3. Their findings suggest that the priming and the route of the vaccine are important for Th-skewing of the responses as well as heterologous protection in mice.

Points to look into:

1. The author claims the strain-specific immunity diminishes rapidly, line 32, is that so and maybe then add some ref here? Or is the efficacy of the vaccine declining due to new escape variants of the virus?
2. There is also lacking ref for the sentence line 48-49; "Protein-based subunit vaccine typically evoke strong antibody responses but poor cellular responses".
3. Fig. 1. Could be improved by having an introduction of how the vaccination has been done, clarifying when the blood samples are taken and number of vaccinations. There is no description of the weeks between the prime boost in the figure, so the blood samples from single vaccination is probably from 3 weeks before the prime-boost samples. The line 84 is adding a shortening: Wks, weeks, not used?
4. Suppl. Fig. 1E, left, the colours is swapped compared to right side, and is the right side showing the individual mice?
5. Line 116, referring to Fig. S3A, but should maybe refer to S2A?
6. Unclear in figures is showing binding to recombinant HA proteins, including both head and stem, or inactivated virus? How will the binding be affected by HA as trimers on virus surface and maybe affected by inactivation and to monomeric HA used in ELISA? This should be discussed.
7. Fig. 2 gives contradictory results in isotype of IgG correlates with cross-reactivity when IM prime IN boost gives better cross-reactivity and IgG2a then for IgG1 after IN – IN as in Fig. 1. (lines 127-129).
8. In Fig. 2, the weight curves and body weight AUC suggest the i.m. prime and i.n. boost is having

highest cross-reactivity, although the IgG nor IgG1 titers suggest that. By increasing the viral load under challenge or reducing vaccine dose, would the survival curves confirm the superior of the i.m. prime and i.n. boost? I would also include the i.m. prime-i.m. boost. Cross-reactive responses is not only from cross-reactive IgG.

9. Fig. 3, the i.m. prime-i.m. boost. group is lacking. The induction of IgG2a systemic and maybe effect of ADCC for cross-reactivity is not thoroughly examined since this group is lacking.

10. Fig. 3 H, I and Fig. S3 B and C shows similar levels of CD4/8+CD127- cells, however, is the naïve mice the best control showing a reduction of CD127+ for as stated in line 220-222 "indicating the transition of more long-lived memory CD8+ T cells into effector T cells"?

11. Fig. 4 A, B and Fig. S4. Highly important with SIgA on mucosal surfaces, however maybe also interesting to look into IgG and isotypes to look at importance of priming for Th skewing also here? Fig. S6

12. Line 331-332, state: "while no significant difference was observed between IM (mRNA)+IN (PHC) and naïve group (Figs. 5G-5I and S7G)". Here is probably the "IM (mRNA)+ IM (mRNA)" correct?

13. In Fig 5. The I.N.(mRNA) + I.N.(mRNA) group is not included here. This regiment is shown to induce higher amount of SIgA than IM (PHC) + IN(mRNA). To reveal the effect of the Trm for cross-reactivity, I can see why the I.N.(mRNA) + I.N.(mRNA) group is left out. It would be interesting to know if IN(Ph3C)+ IN(Ph3C) induced lower amount of the Trm in the mucosa, since the levels of various isotypes did not differ, and if this is due to the mRNA vaccine?

14. They do not confirm that the Th skewing for IgG isotypes is serum is similar on the mucosae and if skewing is due to route of vaccination or the vaccine platform used, the mRNA versus protein/CpG.

15. Line 339-341 is not referring to a figure or ref.

16. A discussion of if the systemic IgG2a also can be a correlation of cross-reactivity in addition to cellular activity can be included.

17. The importance of the prime in systemic Th skewing, if that is due to the route of vaccination (IM versus IN) or due to the vaccine format (mRNA versus protein/CpG) should also be discussed.

Point-by-Point Responses to Reviewer 1's Comments

In this manuscript, Dong et al. investigated multiple homologous and heterologous sequential vaccination strategies in cross-reactive immunity against influenza. The authors demonstrated that 1) vaccine-induced immune responses exhibited Th1 and Th2 biases following mRNA-LNP and CpG-adjuvanted protein nanoparticle vaccine immunization, respectively, 2) intranasal (IN) route of immunization induced stronger mucosal immunity and cross-reactive protection than intramuscular (IM) route, and 3) IM mRNA-LNP priming followed by IN protein nanoparticle boosting afforded optimal cross-reactive infection. The manuscript provided interesting insights. The authors should be applauded for their comprehensive efforts in comparing the effect of immunization routes, vaccine platforms, and immunization orders on vaccine-induced influenza immunity. The ELISA data, as well as the viral protection data, were compelling. The analysis of cross-reactive immunity using heterologous and heterosubtypic flu strains was thorough. And the manuscript was well written and clearly presented. However, several issues reduced my enthusiasm for the paper (see below):

Authors' Response: We sincerely appreciate the reviewer's encouraging feedback, invaluable insights, and thoughtful suggestions. Each comment has significantly contributed to the improvement of our manuscript by strengthening its novelty, offering additional clarity, and providing more comprehensive analyses.

1. Similar strategies utilizing IM mRNA-LNP prime followed by IN boost with protein or adenoviral vector vaccines have been reported previously (Lapiente et al., Nature Communications, 2021; Mao et al., Science, 2022). In addition, the exact formulations of the vaccines have also been reported by the authors, reducing the novelty of the manuscript. While the authors have acknowledged these points in the paper, they should extensively highlight the substantial advancement this manuscript has made compared to previous studies, such as their creative exploration of different immunization combinations, the use of a different viral pathogen as well as the focus on the protective effects on heterologous and heterosubtypic strains.

Authors' Response: We are grateful for the reviewer's insightful suggestions. As suggested, we have highlighted the advancement our manuscript represents compared to previous studies, thereby enhancing its novelty. Please refer to the revisions on **Lines 64-68** highlighted in red in the revised manuscript.

2. Throughout the manuscript, the authors have compared the immunogenicity of IN PHC vs IM mRNA (either as a prime or a boost). An important conclusion that the authors made was that IM mRNA vs IN protein nanoparticle vaccinations induce distinct Th1/Th2 bias and isotype skewing. Several caveats to consider regarding this comparison:

a. Direct comparison between mRNA and protein immunization warrants precaution since the antigen dose could not be controlled (5 ug mRNA may not be "translated" to 5 ug protein). Importantly, the authors showed that IM mRNA priming and IN PHC priming induced comparable titers of total H3 IgG (Fig S1B), suggesting that the antigen doses between the two vaccine platforms were similar. These points should be discussed in the manuscript.

Authors' Response: We agree with the reviewer's rigorous thoughts on comparing the immunogenicity of different vaccine platforms. Initially, we conducted a preliminary study to assess antibody responses and were encouraged to find that both vaccines induced comparable IgG levels. This finding motivated us to comprehensively compare these two vaccines, as detailed in our manuscript. As suggested, we have incorporated relevant discussions to emphasize this point, on **Lines 91 and 360-362** highlighted in red in the revised manuscript. Thank you for helping us clarify this important aspect.

b. Adjuvants play an important part in shaping the antibody isotypes. For PHC vaccines, have the authors examined adjuvants other than CpG? In other words, is Th2 skewing an intrinsic property of PHC immunization or is it also dependent on adjuvant types?

Authors' Response:

We agree that adjuvants play an important role in shaping antibody isotypes. The tendency towards Th2-skewing is an intrinsic property of HA protein vaccines, especially when adjuvants are absent. HA alone triggered Th2-leaning antibody-mediated immune responses and weak cellular responses. Adjuvants may have diverse effects on the Th bias of the immune responses. Alum facilitates Th2-leaning responses, whereas CpG, cGAMP, and MPLA can promote Th1-skewing responses. We developed the PHC (PEI-HA/CpG) nanoparticle in our previous studies [1]. Incorporating CpG can boost IgG2a production and improve the Th balance. However, the overall immune responses still exhibit a Th2-skewing tendency (IgG1>IgG2a).

[1] Polycationic HA/CpG nanoparticles induce cross-protective influenza immunity in mice. *ACS Appl. Mater. Interfaces.*, 2022, 14(5), 6331-6342.

c. Additionally, have the authors looked at IM PHC immunization? Are the immune responses similarly Th2 biased?

Authors' Response: We compared the antibody responses between IN PHC and IM PHC immunization in **Sup. Fig. 8**, and further analyzed the IgG1/IgG2a ratios (**Sup. Fig. 8e**). The results indicated that IM PHC immunization also induced a Th2-biased response, which was not significantly different from IN PHC, suggesting that the immunization route did not show an apparent influence on the Th bias. Related descriptions and further discussions on this observation have been added on **Lines 316-320 and 385-388** in the revised manuscript to clarify the effect of different administration routes.

3. The methodologies used in this manuscript to assess mucosal immunity were flawed, which made data interpretation difficult:

a. The lung tissues are richly vascularized. This means that immune responses measured in the lung would be heavily "contaminated" with immune cells from the circulation if circulating cells are not distinguished (via intravenous CD45 labeling) or removed (through perfusion) from tissue-resident cells. Without a clear way of separating the two compartments, conclusions about mucosal cellular immunity cannot be reliably drawn. This applies to measurement of both mucosal T cells and B cells.

Authors' Response:

We value the reviewer's insightful suggestions. In this study, we did not employ intravenous CD45 labeling techniques; instead, we utilized CD69 as a marker for tissue-resident cells. CD69, in conjunction with CD44, is useful for studying mouse T_{RM} [1-2]. CD69 is a critical distinguishing cell-surface marker that is constitutively expressed on T_{RM} cells in mouse lung tissues and functions as a critical antagonist of S1PR1 (CD363) activity [1-2]. Under steady-state conditions, most T_{RM} expresses CD69. The CD69+ T_{RM} population is phenotypically and transcriptionally distinct from recirculating CD69- memory T cells in both tissues and blood, each characterized by a unique gene expression signature that includes molecules associated with adhesion, migration, and regulation [3-4]. In the revised manuscript, we have provided related descriptions and references on **Lines 263-266**. In the future, we may employ both intravenous CD45 labeling and CD69 labeling techniques to characterize the T_{RM} populations more precisely.

- [1] SnapShot: resident memory T cells. *Cell*, 2014, 157(6), 1488-1488.
- [2] The collagen binding alpha1beta1 integrin VLA-1 regulates CD8 T cell-mediated immune protection against heterologous influenza infection. *Immunity*, 2004, 20(2), 167-179.
- [3] Armed and Ready: Transcriptional Regulation of Tissue-Resident Memory CD8 T Cells. *Frontiers in immunology*, 2018, 9, 406206.
- [4] Human Tissue-Resident Memory T Cells Are Defined by Core Transcriptional and Functional Signatures in Lymphoid and Mucosal Sites. *Cell Reports*, 2017, 20(12), 2921-2934.

b. In addition, all mucosal cellular responses measured in the manuscript were polyclonal which made the data not particularly informative. Tetramer-based approaches should be used to identify antigen-specific T cells and B cells. Alternatively, ELISPOT experiments should be done for lung tissues if tetramer reagents are not available.

Authors' Response: We appreciate the reviewer's suggestions. The primary objective of this study is to investigate the influence of various immunization strategies, routes, and sequences on the induction of antigen-specific immune responses. Consequently, we focused our analysis on cytokine-secreting T cells and antibody-secreting B cells specific to the entire antigens in our study. Although a more detailed investigation into epitope-specific responses would yield valuable insights into antigen epitopes, it falls outside the scope of our current study. Nonetheless, this aspect certainly merits further investigation, and we may consider expanding the scope of our research to include this in future studies.

c. Absolute cell count (# of antigen-experience CD4 and CD8 T cells in the lung and airways) should be included in addition to representative flow plots and relative percentages.

Authors' Response:

In this study, we did not count the cell numbers. The immune cell numbers in the airways, particularly in naïve or IM-immunized mice, are comparatively low. Our experiment revealed that IN-immunized mice have significantly higher immune cells in the BALF than their naïve counterparts. Therefore, during FACS, we used all naïve BALF samples but only a portion from immunized mice. If we were to count the total CD44+CD4+ T or CD44+CD8+ cells, the difference between groups would be more evident.

Moreover, the absolute cell counts can vary based on the processing protocols and techniques, leading to significant fluctuations. The relative cell percentages provide a more standardized and predictive measure of immune status. Unlike absolute cell counts, relative cell percentages are not influenced by sample variances or differences in sample collection and processing methods. The relative percentages consider both the total number of lymphocytes and the specific population simultaneously, offering a more reliable parameter for analysis. For the flow cytometry analysis of lung cells, we typically set a threshold of 10^4 for CD8 T cells or 2×10^4 for CD4 T cells. This approach ensures consistency and reliability in our data. In future studies, we will consider both absolute cell counts and relative percentages in our analyses to provide a more comprehensive assessment of immune responses.

4. To further strengthen the serology data, serum and mucosal neutralization against homologous and heterologous influenza strains should be provided for each immunization strategies. In addition, mucosal IgG titer in the BALF and nasal washes should be measured. Do mucosal IgG titers track with protection?

Authors' response:

In response to the reviewer's suggestion, we performed the serum microneutralization assay against Phi and rSH and did not detect any discernible neutralization activity. We have included the results and corresponding descriptions and discussions in the revised manuscript (**Sup. Fig.2e-f, Sup. Fig.6g-k, Lines 146-147, 159, 293-294, 299-300, 381**).

It has been reported that sIgA dominates the antibody response in the upper respiratory tract. In contrast, IgG is the major antibody isotype in the lower respiratory tract. Therefore, we evaluated antigen-specific IgG profiles in BALF. We observed comparable levels of H3- and Phi-specific IgG in BALF in groups including IM (mRNA)+IN (PHC), IN (PHC)+IM (mRNA), IN (PHC)+IN (PHC), and IM (mRNA)+IM (PHC). Our findings suggest that BALF IgG profiles mirror those in the immune sera, consistent with previous reports [1], and are not the primary correlate of cross-protection. Moreover, no apparent neutralization activity against either Phi or rSH was observed. We have incorporated these results, descriptions, and discussions into the revised manuscript (**Sup. Fig.4g-k, Sup. Fig.6e-g,k, Lines 238-247, 290-294, 425-426**).

Besides, our primary focus is on cross-protection rather than homologous protection. It is common for these vaccines to generate homologous neutralizing antibodies. We demonstrated robust HAI activity in the sera and BALF of PHC-immunized mice against homologous Aic in the earlier study [2]. We did not include nasal washes in this assay because of limited nasal sample volumes.

[1] The human antibody response to influenza A virus infection and vaccination. *Nat. Rev. Immunol.* 2022, 19, 383-397.

[2] Polycationic HA/CpG nanoparticles induce cross-protective influenza immunity in mice. *ACS Appl. Mater. Interfaces.*, 2022, 14(5), 6331-6342.

5. A thorough discussion of why the heterologous sequential immunization strategy led to better cross-protection would further strengthen the manuscript.

Authors' response: We appreciate the reviewer's insightful feedback on our manuscript. As suggested, we have added a paragraph to discuss the rationale and advantages of the heterologous sequential immunization strategy for influenza cross-protection on **Lines 466-477** to strengthen the manuscript.

Below I listed additional minor issues and comments I have that if addressed could further improve the quality of the manuscript.

6. In Line 91, besides hydrodynamic sizes (Fig S1A), polydispersity index (PDI) of PHC and LNP nanoparticles should be reported as well.

Authors' Response: Following the reviewer's suggestion, we have provided the PDI data in **Sup. Fig. 1a** and added a related description on **Line 87** of the revised manuscript.

7. In Line 119, the findings that IgG1 leads to improved antibody cross-reactivity is interesting. Can the authors provide some potential mechanisms?

Authors' Response:

We appreciate the reviewer's valuable suggestions. This finding exceeded our initial expectations. We used the same serum batch for antibody analysis against diverse viruses or HAs and found this phenomenon. Our results suggest a potential correlation between IgG subtype and cross-reactivity, which seems to depend on the antigenic divergence distance. Notably, a dominant IgG1 antibody profile was observed for the distantly related heterosubtypic rSH or H7. This finding is interesting. However, we have not identified a mechanism that can explain this point.

Upon reviewing online publications, we found a recent study reporting correlations between the IgG subclass and antibody breadth, potentially driven by differences in hinge flexibility among the IgG subclasses [1]. However, some of these antibodies may be strain-specific, whereas others may recognize multiple strains. It has been reported that each clone exhibits unique cross-reactivity and virus-neutralization capacity [2]. Determining how this occurs is interesting. However, this falls outside the scope of our present study.

As these antibodies did not directly correlate with the cross-protection we observed, we did not focus on them in the present study. We have suggested a potential correlation and emphasized the need for additional evidence and further comprehensive studies to verify this correlation or to explain the observed phenomena (**Lines 124, 391, and 399**).

[1] IgG3 subclass antibodies recognize antigenically drifted influenza viruses and SARS-CoV-2 variants through efficient bivalent binding. *Proceedings of the National Academy of Sciences*, 2023, 120(35), e2216521120.

[2] Discriminating cross-reactivity in polyclonal IgG1 responses against SARS-CoV-2 variants of concern. *Nature Communications*, 2022, 13(1), 6103.

8. In Line 202, the authors performed ELISPOT assay to detect ASC response in the spleen and bone marrow. The bone marrow data are important and should be moved from Fig S3 in the main Figures.

Authors' response: As suggested, we have moved the bone marrow ASC data to Fig. 3 and updated related descriptions in the revised manuscript.

9. In Line 446, results shown in Fig S9 should be described in the Main Text, not just in the Discussion section.

Authors' response: We appreciate the reviewer's valuable suggestions. As suggested, we have added a separate section on **Lines 327-345** to describe Sup. Fig. 9 in the revised manuscript.

10. In Fig S7, error bars are missing.

Authors' response: We used pooled BALF samples for the Naïve and IM (mRNA)+IM (PHC) groups, and we have specified this information in the **Sup. Fig. 7 legend** to ensure clarity. We started flow cytometry analysis on BALF samples from the Naïve and IM (mRNA)+IM (PHC) groups using a compensation setup that works well for the CD45+ population but is suboptimal for CD4/CD8 separations. Consequently, we refined the compensation setup and completed the analysis. Given the limited BALF sample availability and the inherently lower number of immune cells in these two groups, we pooled cells from three mice for FACS analysis. Only data obtained from the refined compensation setup were included in the manuscript.

Point-by-Point Responses to Reviewer 2's Comments

The authors have investigated the effect of a prime – boost regiment with intra nasal mRNA lipid nanoparticle and intra muscular protein-based PHC nanoparticle vaccines targeting influenza hemagglutinin, H3. Their findings suggest that the priming and the route of the vaccine are important for Th-skewing of the responses as well as heterologous protection in mice.

Authors' response: We appreciate the reviewer's valuable comments. In this study, we utilized an intramuscular mRNA lipid nanoparticle and an intranasal protein-based PHC nanoparticle. Our findings suggest that the priming vaccine type, rather than the route, is crucial for the Th-skewing of the responses. Additionally, both administration sequences (or priming modality) and routes are important for heterologous protection in mice.

Points to look into:

1. The author claims the strain-specific immunity diminishes rapidly, line 32, is that so and maybe then add some ref here? Or is the efficacy of the vaccine declining due to new escape variants of the virus?

Authors' response: Following the reviewer's suggestion, we have added the related reference **[1] on Line 32** in the revised manuscript. The diminished efficacy of influenza vaccines is mainly due to the antigenic mismatch between circulating viruses and vaccine strains. Influenza viruses continue to evolve through genetic mutation, evading pre-existing immunity. On the other hand, the vaccine-induced immune response, as opposed to natural virus infection, is short-lived—a

point reiterated in many published articles. Dr. Krammer underscored this by stating, “Vaccines are the best available countermeasure against infection, but vaccine effectiveness is low compared with other viral vaccines, and the induced immune response is narrow and short-lived” [1]. We have replaced “diminish rapidly” with “short-lived” to maintain consistency.

[1] Krammer, Florian. The human antibody response to influenza A virus infection and vaccination. *Nature Reviews Immunology*. 2019, 19 (6): 383-397.

2. There is also lacking ref for the sentence line 48-49; “Protein-based subunit vaccine typically evoke strong antibody responses but poor cellular responses”.

Authors’ response: As suggested, we have added the reference [10] on Line 49 in the revised manuscript.

3. Fig. 1. Could be improved by having an introduction of how the vaccination has been done, clarifying when the blood samples are taken and number of vaccinations. There is no description of the weeks between the prime boost in the figure, so the blood samples from single vaccination is probably from 3 weeks before the prime-boost samples. The line 84 is adding a shortening: Wks, weeks, not used?

Authors’ response: We appreciate the reviewer’s valuable suggestions. In response, we added related descriptions in the legend of Fig. 1a to outline the vaccination protocol. We also relocated the Schematic illustration depicting the immunization and sera collection timeline from the Supplementary Information to Fig. 1a. Relevant descriptions were also detailed in the Methods section (Lines 531-539). Additionally, we deleted the unused shortening (Wks, weeks).

4. Suppl. Fig. 1E, left, the colours is swapped compared to right side, and is the right side showing the individual mice?

Authors’ response: We appreciate the reviewers for bringing our oversight to our attention. We have updated Sup. Fig. 1e to ensure that the color labeling aligns with other results and incorporated detailed descriptions in the figure legend.

5. Line 116, referring to Fig. S3A, but should maybe refer to S2A?

Authors’ response: We appreciate the reviewer for pointing out our mistake. It refers to the present Sup. Fig. 2a. The revision have been incorporated into the revised manuscript (Line 112).

6. Unclear in figures is showing binding to recombinant HA proteins, including both head and stem, or inactivated virus? How will the binding be affected by HA as trimers on virus surface and maybe affected by inactivation and to monomeric HA used in ELISA? This should be discussed.

Authors’ response:

We appreciate the reviewer’s questions. We did not use monomeric HA in any tests conducted in this study. Instead, we used GCN4-stabilized trimeric full-length recombinant HA proteins. We have provided further elaboration on the HA to clarify this point in the revised manuscripts (Lines 114, 506-507).

Both the HA proteins and viruses we used consist of HA trimers. We found some differences in the antibody endpoint titers against full-length HAs and inactivated viruses, such as H3 vs. Aic. Generally, higher titers were observed using HA as the coating antigens. Both the HA exposure style and the inactivation process may have some effects. Determining the exact factor responsible for these differences is challenging, and may necessitate more data for thorough verification and explanation. However, a comprehensive comparison between HA and virus-specific responses falls outside the scope of our study.

Regardless, both HA and virus antigens could be used to compare immunization groups. The results consistently displayed a similar trend irrespective of the antigen used. To make our discussion more precise, we specified the antigens used throughout the manuscript. When interpreting ELISA results involving HA, we label the antigen as HA. Alternatively, when utilizing inactivated virus samples for testing, we label them with the corresponding virus name, such as Aic or Phi.

7. Fig. 2 gives contradictory results in isotype of IgG correlates with cross-reactivity when IM prime IN boost gives better cross-reactivity and IgG2a then for IgG1 after IN – IN as in Fig. 1. (lines 127-129).

Authors' response:

We appreciate the reviewer's questions. The results are not contradictory. The correlation between the IgG subtype and cross-reactivity depends on the antigenic divergence distance. Specifically, we found that Th2-skewing groups tend to induce higher cross-reactive antibody responses, particularly against rSH), motivating us to hypothesize a correlation between the IgG subtype and cross-reactivity (**Lines 113-125**). The dominance of IgG1 in H7-specific antibody profiles further supports our hypothesis (**Lines 160-162**). However, this does not imply that only IgG1 is correlated with cross-reactivity or cross-protection.

The IM (mRNA)+IN (PHC) group induced relatively balanced IgG1 and IgG2a antibody profiles against homologous Aic or H3 or heterologous Phi virus. The Phi-specific IgG1 antibody titers were slightly greater than or equal to Phi-specific IgG2a titers (**Fig. 2c-d**). However, there was an IgG1-dominant profile against the far distant heterosubtypic rSH. This result indicated a dependence on antigenic divergence distance. We modified our discussions to indicate that this may represent a possible correlation and emphasized the need for additional evidence and further in-depth studies to confirm this speculation (**Lines 124, 391-399**). To summarize, the IM (mRNA)+IN (PHC) group showed good cross-protection and good IgG 2a (not higher than IgG1). This result is consistent with our claims in the revised manuscript.

8. In Fig. 2, the weight curves and body weight AUC suggest the i.m. prime and i.n. boost is having highest cross-reactivity, although the IgG nor IgG1 titers suggest that. By increasing the viral load under challenge or reducing vaccine dose, would the survival curves confirm the superior of the i.m. prime and i.n. boost? I would also include the i.m. prime-i.m. boost. Cross-reactive responses is not only from cross-reactive IgG.

Authors' response:

We appreciated the reviewer's questions. We agree that cross-reactive responses are not only attributed to cross-reactive IgG. We demonstrated that the i.m. prime and i.n. boost group, IM (mRNA)+IN (PHC), exhibited the highest efficacy of cross-protection (**Fig. 2**). However, we also found that the cross-reactive antibody responses did not consistently correlate with cross-protection, promoting further investigation into cellular and mucosal responses.

In our study, both the virus infection dose and vaccine dose were carefully chosen, ensuring a reasonable balance. Changes in dose do not impact the inter-group comparison. Differences in survival rates between groups may become more evident with a higher viral infection dose or a lower vaccine dose. However, we anticipate that the IM (mRNA)+IN (PHC) group will maintain its superior performance in survival rates and body weight among all groups. With several challenge rounds in our study, the IM (mRNA)+IN (PHC) group consistently showed the robust protection efficacy upon challenges.

Besides, we included the i.m. prime-i.m. boost groups for comparison. We compared the cross-protection efficacies between the IM (mRNA)+IM (PHC) and IM (mRNA)+IN (PHC) and between the IM (PHC)+IM (PHC) and IN (PHC)+IN (PHC) groups (**Fig. 5, Sup. Figs. 6,8**). Please refer to the manuscript for detailed results and discussions on these comparisons.

9. Fig. 3, the i.m. prime-i.m. boost. group is lacking. The induction of IgG2a systemic and maybe effect of ADCC for cross-reactivity is not thoroughly examined since this group is lacking.

Authors' response:

We appreciated the reviewer's questions. We are not sure which specific group the reviewer referred to as the i.m. prime-i.m. boost group. It could be either IM (mRNA) + IM (mRNA), IM (mRNA)+IM (PHC), or IM (PHC)+IM (PHC).

In the context of mouse models, it has been well established that IgG2a antibodies are correlated with ADCC activity [1-2], a point we also verified in our previous study [3]. In the present study, ADCC activity was not a main factor contributing to protection for several reasons:

1. Both our previous [4] and current studies showed that IM (mRNA)+IM (mRNA)-immunized mice displayed high Phi-specific IgG2a titers but experienced severe body weight loss post-Phi infection;
2. Mice from the IM (mRNA)+IM (PHC) group exhibited significantly greater bodyweight loss, compared to those from the IM (mRNA)+IN (PHC) group, despite comparable Phi-specific IgG profiles.
3. ADCC did not contribute to protection against rSH owing to the IgG1-dominant antibody profiles.

Therefore, the ADCC activity-mediated protection is limited in our study and cannot fully explain the difference in protection efficacy among diverse groups. We placed greater emphasis on cellular and mucosal immune responses. However, considering the distinct IgG subtype profile against Phi and rSH, we speculate that ADCC may play some role in heterologous protection against Phi but not in the heterosubtypic protection against rSH. We have incorporated

discussions on the possible effect of ADCC (**Lines 393-402**, highlighted in red) into the revised manuscript.

- [1] The potential role of Fc-receptor functions in the development of a universal influenza vaccine. *Vaccines* 2018, 6(2): 27.
- [2] Influenza-specific antibody-dependent cellular cytotoxicity: toward a universal influenza vaccine. *The Journal of Immunology*, 2014, 193(2), 469-475.
- [3] Polycationic HA/CpG nanoparticles induce cross-protective influenza immunity in mice. *ACS applied materials & interfaces*, 2022, 14(5), 6331-6342.
- [4] cGAMP-adjuvanted multivalent influenza mRNA vaccines induce broadly protective immunity through cutaneous vaccination in mice. *Molecular Therapy-Nucleic Acids*, 2022, 30, 421-437.

10. Fig. 3 H, I and Fig. S3 B and C shows similar levels of CD4/8+CD127- cells, however, is the naïve mice the best control showing a reduction of CD127+ for as stated in line 220-222 “indicating the transition of more long-lived memory CD8+ T cells into effector T cells”?

Authors’ response: We appreciate the reviewer’s questions. Before obtaining the results, we did not specifically select naïve mice as the optimal control. We aimed to identify differences between groups, and we were surprised to observe the significant down-regulation of CD127 expression in all immunized groups. However, significant differences were noted between naïve and immunized mice. Under the circumstances, multiple markers should be used simultaneously to precisely characterize long-lived memory T cells. In the revised manuscript, we have deleted the sentence “indicating the transition of more long-lived memory CD8+ T cells into effector T cells” to enhance precision in our description.

11. Fig. 4 A, B and Fig. S4. Highly important with SIgA on mucosal surfaces, however maybe also interesting to look into IgG and isotypes to look at importance of priming for Th skewing also here? Fig. S6

Authors’ response: We appreciate the reviewer’s valuable comments. We evaluated the antigen (H3 and Phi)-specific IgG subtype profiles and IgG1/IgG2a ratios in BALF, which aligned with our findings from immune sera. **Sup. Fig. 4g-h** and the corresponding discussion (**Lines 238-247, 425**) have been incorporated into the revised manuscript to provide additional context.

12. Line 331-332, state: “while no significant difference was observed between IM (mRNA)+IN (PHC) and naïve group (Figs. 5G-5I and S7G)”. Here is probably the “IM (mRNA)+ IM (mRNA)” correct?

Authors’ response: We appreciate the reviewer for pointing out our mistake. It should be “IM (mRNA)+IM (PHC)”. The corresponding change has been made in **Line 310** in the revised manuscript.

13. In Fig 5. The I.N.(mRNA) + I.N.(mRNA) group is not included here. This regiment is shown to induce higher amount of SIgA than IM (PHC) + IN(mRNA). To reveal the effect of the Trm for cross-reactivity, I can see why the I.N.(mRNA) + I.N.(mRNA) group is left out. It would be

interesting to know if IN(Ph3C)+ IN(Ph3C) induced lower amount of the Trm in the mucosa, since the levels of various isotypes did not differ, and if this is due to the mRNA vaccine?

Authors' response:

We appreciate the reviewer's questions. In this study, we did not include the IM (PHC)+IN(mRNA) group due to the absence of a proper mRNA vaccine formulation for the intranasal route. Both our mRNA-LNP vaccine and commercial mRNA-LNP vaccines (e.g., current COVID-19 mRNA vaccine) are designed for intramuscular injection, as they cannot be safely administered intranasally due to lipid toxicity concerns. Our study focuses on two vaccines: an mRNA-LNP vaccine intended solely for intramuscular use and a protein-based PHC vaccine suitable for both IM and IN routes. Using these vaccines—IM (mRNA), IN (PHC), and IM (PHC)—we aim to provide comprehensive insights into the optimization of vaccination strategies.

Besides, we have shown the Trm results of the IN(PHC)+ IN(PHC) group in **Fig. 4d-e**. Our findings indicate that the Trm induction mainly depends on the immunization route rather than the vaccine type. We have added related descriptions and discussions on **Lines 266-269 and 427-430** in the manuscript.

14. They do not confirm that the Th skewing for IgG isotypes in serum is similar on the mucosae and if skewing is due to route of vaccination or the vaccine platform used, the mRNA versus protein/CpG.

Authors' response:

We appreciated the reviewer's questions. We evaluated antigen-specific IgG profiles in BALF and observed similar results to those in immune sera. We have added related results (**Sup. Fig. 4g-h**) and discussion (**Lines 238-247, 425**) in the revised manuscript. Please refer to our **responses to Question 11**.

To analyze the main reason for the Th skewing, we investigated the influence of vaccination routes (IM versus IN) on the Th skewing. We analyzed the IgG1/IgG2a ratios in **Sup. Figs. 6,8** to compare the Th bias between the IM (mRNA)+IM (PHC) and IM (mRNA)+IN (PHC) and between the IM (PHC)+IM (PHC) and IN (PHC)+IN (PHC) groups. With the same PHC vaccines, the variation of immunization route (IM or IN) did not significantly change the IgG1/IgG2a ratio. Based on our results, we concluded that the IgG Th skewing is determined by the vaccine platform used for priming immunization rather than the route of vaccination.

We have added related descriptions and discussions and provided potential mechanisms to clarify the influence of the administration routes in the revised manuscript (**Lines 291-293, 317-320, 385-388**, highlighted in red).

15. Lines 339-341 is not referring to a figure or ref.

Authors' response: As suggested, we have added the referring figure (**Fig. 5j, Sup. Fig. 8a**) on **Line 322** of the revised manuscript.

16. A discussion of if the systemic IgG2a also can be a correlation of cross-reactivity in addition to cellular activity can be included.

Authors' response: We appreciated the reviewer's questions. The systemic IgG2a also correlates with cross-protection. We have discussed this point in the manuscript (**Lines 393-402**). Considering the IgG subtype profile in different groups, it may play some roles in the cross-protection against heterologous Phi, but not against rSH. Please refer to our **answers to Question 9** for more information.

17. The importance of the prime in systemic Th skewing, if that is due to the route of vaccination (IM versus IN) or due to the vaccine format (mRNA versus protein/CpG) should also be discussed.

Authors' response: We appreciated the reviewer's questions. We have added more analyses on this point. Based on our results, we concluded that the IgG Th skewing is determined by the vaccine platform used for priming immunization rather than the route of vaccination. Please refer to our detailed **responses to Question 14** for more information.

Reviewers' Comments:

Reviewer #1:

Remarks to the Author:

The authors have comprehensively addressed the reviewer's comments. The manuscript is now much clearer and the additional data significantly added to the overall message - I have no further comments. I congratulate the authors on their excellent work.

Reviewer #2:

Remarks to the Author:

The authors have addressed my previous general and specific comments regarding the initial submission of the manuscript, including the requested additional experiments and clarifications that have helped to improve the paper.